# Small-scale proxies for large-scale Transformer training instabilities

**Mitchell Wortsman**[†]  **Peter J. Liu**  **Lechao Xiao**  **Katie Everett**
**Alex Alemi**  **Ben Adlam**  **John D. Co-Reyes**  **Izzeddin Gur**  **Abhishek Kumar**
**Roman Novak**  **Jeffrey Pennington**  **Jascha Sohl-dickstein**[†]  **Kelvin Xu**
**Jaehoon Lee**[*]  **Justin Gilmer**[*]  **Simon Kornblith**[*†]

Google DeepMind

## ABSTRACT

Teams that have trained large Transformer-based models have reported training instabilities at large scale that did not appear when training with the same hyperparameters at smaller scales. Although the causes of such instabilities are of scientific interest, the amount of resources required to reproduce them has made investigation difficult. In this work, we seek ways to reproduce and study training instability at smaller scales. First, we focus on two sources of training instability described in previous work: the growth of logits in attention layers (Dehghani et al., 2023) and divergence of the output logits from the log probabilities (Chowdhery et al., 2022). By measuring the relationship between learning rate and loss across scales, we show that these instabilities also appear in small models when training at high learning rates, and that mitigations previously employed at large scales are equally effective in this regime. This prompts us to investigate the extent to which other known optimizer and model interventions influence the sensitivity of the final loss to changes in the learning rate. To this end, we study methods such as warm-up, weight decay, and the $\mu$Param (Yang et al., 2022), and combine techniques to train small models that achieve similar losses across orders of magnitude of learning rate variation. Finally, to conclude our exploration we study two cases where instabilities can be predicted before they emerge by examining the scaling behavior of model activation and gradient norms.

## 1  INTRODUCTION

Scaling up transformers has led to remarkable progress from chat models to image generation. However, not every training run is successful. When training large Transformers, researchers have reported instabilities which slow or destabilize learning (Chowdhery et al., 2022; Dehghani et al., 2023; Zhang et al., 2022; Molybog et al., 2023; Cohen et al., 2022). As the resources required for large runs continue to grow, it is important to examine the ways that Transformer training can fail.

In this report we reproduce, study, and predict training instability in Transformer models. We find that measuring the relationship between learning rate and loss across scales is a useful tool to identify instability (e.g., Figure 1). Therefore, we introduce learning rate (LR) sensitivity, which serves as a useful summary statistic for learning rate vs. loss curves. LR sensitivity measures the deviation from optimal performance when varying LR across orders of magnitude.

We show that two sources of instability, which have previously been described at scale, can be reproduced in small Transformers.[1] This enables their study without access to large resource pools. In particular, we examine the growth of logits in attention layers (Dehghani et al., 2023; Gilmer et al.; Zhai et al., 2023a) and divergence of the output logits from the log probabilities (Chowdhery et al., 2022). As evident from the learning rate vs. loss curves and by inspecting model characteristics, both instabilities appear at high learning rates in small models. Moreover, interventions which have previously been employed at scale are also successful in this regime (e.g., Figure 1). These

---

[*]Equal contribution.  [†]Authors are now at Anthropic.
[1]We focus on instabilities which lead to slow divergence, not loss spikes (see Section 4).

interventions—qk-layernorm (Dehghani et al., 2023) and z-loss regularization (Chowdhery et al., 2022)—reduce LR sensitivity and enable successful training across three orders of magnitude of learning rate variation.

These observations raise the question of how other known optimizer and model interventions affect the shape of the learning rate vs. loss curves across scales. Therefore, we study the effect of techniques such as warm-up, weight decay, and $\mu$Param (Yang et al., 2022) in this context. When employing qk-layernorm and z-loss regularization, these other techniques usually have little impact on the range of learning rates at which models can be stably trained, but do affect the sensitivity to learning rate within this range. In line with previous work, we find that longer warm-up reduces learning rate sensitivity, as does the independent scaling of learning rate and weight decay recommended by Loshchilov & Hutter (2019). One interesting finding is that scaling depth increases LR sensitivity at a faster rate than scaling width.

The remainder of our investigation centers on the scaling behavior for model characteristics such as activation and gradient norms. Using the attention logit growth instability as an example, we show that it is possible to predict an instability before it emerges. This is in contrast to prior works on scaling which primarily focus on scaling trends related to loss (Kaplan et al., 2020; Hoffmann et al., 2022).

We conclude by using the scaling behavior of model characteristics to search for instabilities that are currently not well documented. Our investigation shows that gradient norms decrease with both scale and learning rate, such that the default AdamW (Loshchilov & Hutter, 2019) epsilon hyperparameter is too large. This causes updates that are too small. We connect this phenomenon and the attention logit growth instability to parameter norm growth (Merrill et al., 2020; Lee, 2023).

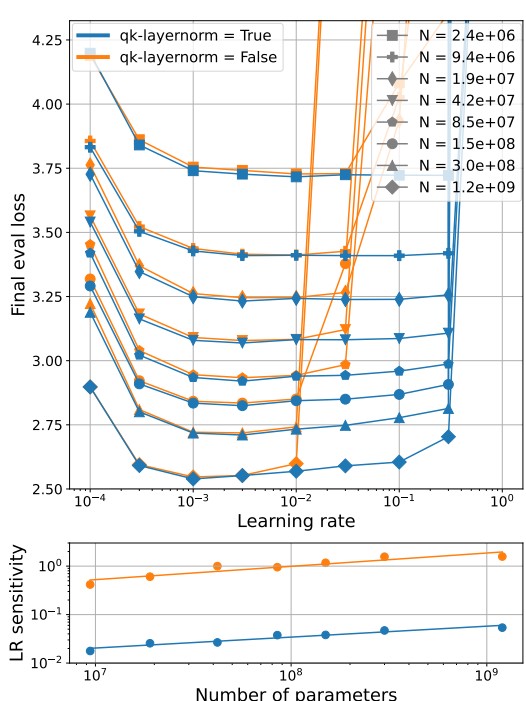

Figure 1: Qk-layernorm (Dehghani et al., 2023) enables stable training across three orders of magnitude of learning rate (LR) variation. **(Top)** For transformers with $N$ parameters, we plot the effect of learning rate on final evaluation loss. **(Bottom)** We use LR sensitivity to summarize the top plot. LR sensitivity measures the expected deviation from the minimum achieved loss when varying learning rate across three orders of magnitude. Qk-layernorm reduces LR sensitivity, but LR sensitivity still increases with model scale.

Overall, we believe our work presents new scientific opportunities for studying training stability without access to large resource pools.

## 2 EXPERIMENTAL METHODOLOGY

This section details our experimental set-up (Section 2.1) and useful tools employed by our analysis: (i) measuring the relationship between learning rate and loss across scales (Section 2.2) and (ii) examining scaling trends for model characteristics (Section 2.3).

### 2.1 EXPERIMENTAL SET-UP

We train small Transformer models (Vaswani et al., 2017) with a similar experimental set-up as GPT-2 (Radford et al., 2019) implemented in Flax (Heek et al., 2023): the models are decoder-only (Liu et al., 2018) and trained with an auto-regressive loss (refer to Section A for more infrastructure

details). While we experimentally manipulate many of the following hyperparameters, this section provides their default values, which we use unless otherwise specified.

By default, we use AdamW (Loshchilov & Hutter, 2019) with $\beta_1 = 0.9$, $\beta_2 = 0.95$, $\epsilon = $ 1e-8, and gradient clipping at global norm 1. The default warmup is 5e3 steps, and the default number of total steps is 1e5. We use a linear schedule for warmup and and a cosine-decay (Loshchilov & Hutter, 2016) schedule for the remainder, with minimum learning rate 1e-5. We use an independent weight decay of 1e-4 and auxiliary z-loss (Chowdhery et al., 2022) with coefficient 1e-4. Sections 3.2.2 and 3.1.2 respectively provide additional information and ablations on decoupled weight decay and z-loss. We use pre-normalization (Radford et al., 2019) Transformers with qk-layernorm (Dehghani et al., 2023) (see Section 3.1.1 for information). We do not use any biases following Chowdhery et al. (2022), and the layernorm (Ba et al., 2016) $\epsilon$ remains at the default value in Flax (Heek et al., 2023) of 1e-6. We jointly scale up the embedding size, depth, and number of heads when scaling parameters. We do not use weight tying of the first and last layer (Press & Wolf, 2017), and when reporting the number of parameters we exclude the embedding and head (as in Kaplan et al. (2020)). We use rotary positional embeddings (Su et al., 2021), and for training data we use C4 (Raffel et al., 2020a). Letting $d$ refer to the model dimension (i.e., the embedding size), the feed-forward component of the Transformer is an MLP with hidden dimension of $4d$ and gelu (Hendrycks & Gimpel, 2016) activations. As in Vaswani et al. (2017) we use factor $1/\sqrt{d}$ scaling in the self-attention. The embedding initialization is the default in Flax, which is normally distributed with standard deviation $1/\sqrt{d}$. The remainder of the weights are initialized with a truncated normal distribution with inverse root fan-in standard deviation (Glorot & Bengio, 2010). The default batch size is 256, where each batch element has a sequence length of 512 tokens. Sequences are packed so that no padding is required. Finally, we use the vocabulary from Raffel et al. (2020b) which has size 32101 and uses a SentencePiece (Kudo & Richardson, 2018) tokenizer. We train on TPUs (Jouppi et al., 2017) in bfloat16 precision using Flax (Heek et al., 2023) and JAX (Bradbury et al., 2018).

## 2.2 LR VS. LOSS CURVES AND LEARNING RATE SENSITIVITY

To investigate how model instability emerges with scale, it is useful to plot the relationship between learning rate (LR) and loss for models of different sizes. For instance, an instability is often characterized by an explosion in the loss at high learning rates. LR vs. loss curves can reveal how the lowest unstable learning rate changes as a function of model size.

To summarize LR vs. loss curves, we use LR sensitivity. LR sensitivity measures the deviation in final validation loss from optimal when sweeping LR across three orders of magnitude. If a model fails to train at high learning rates, then LR sensitivity will be high. There are cases where LR vs. loss curves and LR sensitivity are no longer meaningful, for instance if an intervention changes the meaning of learning rate—see Appendix B for a detailed discussion.

Let $\theta = \mathcal{A}(\eta)$ denote the model weights $\theta$ obtained when training with learning rate $\eta$, and let $\ell(\theta)$ denote the validation loss when using weights $\theta$. For a learning rate range $[a, b]$, let $\ell^*$ denote the loss obtained with the best learning rate, i.e., $\ell^* = \min_{\eta \in [a,b]} \ell(\mathcal{A}(\eta))$. Moreover, let $\ell_0$ denote loss at initialization. Then, LR sensitivity is defined as $\mathbb{E}_{\eta \in [a,b]} [\min (\ell (\mathcal{A}(\eta)), \ell_0) - \ell^*]$.

Unless otherwise mentioned, we use the learning rate range 3e-4 to 3e-1 with AdamW (Loshchilov & Hutter, 2019) to measure LR sensitivity, where LR refers to the maximum value in a cosine decay schedule with warm-up (Loshchilov & Hutter, 2016). We consider LRs in {3e-4, 1e-3, 3e-3, 1e-2, 3e-2, 1e-1, 3e-1} when computing the minimum and expectation.

## 2.3 SCALING TRENDS FOR MODEL CHARACTERISTICS

To study instability, we also find it useful to examine scaling trends for model characteristics such as gradient or activation norms. This method is helpful for predicting instabilities and contrasts with previous work on scaling, which primarily focuses on trends relating model scale and loss (Kaplan et al., 2020; Hoffmann et al., 2022).

## 3 RESULTS

This section presents our results on training stability for small Transformers. Equipped with LR sensitivity (Section 2.2), we study two known instabilities and their corresponding mitigation at

small scale (Section 3.1). This raises the question of how other model and optimizer interventions effect sensitivity of final loss to learning rate, which we investigate in Section 3.2. Finally, we examine whether instabilities can be reliably predicted before they emerge: Section 3.3 predicts when the logit growth instability may cause divergence in a larger model, while Section 3.4 aims to find other issues that may occur when scaling up with our default hyperparameters.

## 3.1 REPRODUCING TWO KNOWN INSTABILITIES AT SMALL SCALE

Here, we examine two instabilities that have previously been described at scale: the growth of logits in attention layers (Dehghani et al., 2023; Gilmer et al.; Zhai et al., 2023a) and divergence of the output logits from the log probabilities (Chowdhery et al., 2022). By examining LR vs. loss curves, we show that these instabilities can be reproduced in small models by using high learning rates and that mitigations employed at scale are effective in this regime.

### 3.1.1 ATTENTION LOGIT GROWTH

Researchers have previously documented that Transformer training fails when the attention logits become large (Dehghani et al., 2023; Zhai et al., 2023a). In Dehghani et al. (2023), this issue emerged when training a ViT model (Dosovitskiy et al., 2021) with 22 billion parameters.

In the self-attention layer of a Transformer (Vaswani et al., 2017), queries $q_i$ and keys $k_i$ are combined to compute the attention logits $z_{ij} = \langle q_i, k_j \rangle / \sqrt{d_h}$, where $d_h$ is the head dimension. Next, the attention logits are passed through a softmax to produce attention weights, which are used to combine values $v_i$. Dehghani et al. (2023) observed that the attention logits $z$ became large, which they refered to as attention logit growth. As a result, the attention weights collapse to one-hot vectors, which was named attention entropy collapse by Zhai et al. (2023a). To resolve this issue, Dehghani et al. (2023) proposed qk-layernorm, which applies LayerNorm (Ba et al., 2016) to the queries and keys before computing the attention logits.

In our experiments, we find that models need not be large to exhibit instability related to attention logit growth. As shown in Figure 1, the maximum learning rate at which small models can be trained increases when using qk-layernorm. Without qk-layernorm, the learning rate at which models diverge becomes smaller with increasing model size. By contrast, models with qk-layernorm exhibit considerably lower LR sensitivity and train to low loss at high learning rates. As a highlight, qk-layernorm allows training a model with 1.2B parameters at learning rate 0.3. Both with and without qk-layernorm, LR sensitivity increases with scale.

Figure G.1 displays the loss and max attention logit for two model scales that differ by three orders of magnitude. In both cases, the loss diverges without qk-layernorm. Our results in Appendix Figure G.7 suggest that attention logit growth is due to growth in the queries and keys, not due to an increase in their alignment. Finally, Appendix C connects this instability to the quadratic dependence of attention logits on parameter norms.

### 3.1.2 OUTPUT LOGIT DIVERGENCE

Another instability reported by researchers training large models is divergence in the output logits from the log probabilities (Chowdhery et al., 2022). Just as before, we reproduce this instability with small models at large learning rates, and the proposed mitigation ameliorates the issue. Overall, Figure 2 summarizes the effect.

Let $y$ denote the model's output logits, which are used to compute class probabilities $p_i$ via a softmax $p_i = e^{y_i}/Z$ where $Z = \sum_j e^{y_j}$. This instability occurs when the logits diverge and become very negative, as illustrated in Figure G.2 for a 2.4M parameter model at learning rate 0.1. In contrast to the attention logit growth instability, this divergence occurs towards the end of training. The mitigation proposed by Chowdhery et al. (2022) is to encourage $\log Z$ to remain close to zero. They add an auxiliary loss $\log^2 Z$, referred to as z-loss, with coefficient 1e-4.

As illustrated in Figures 2, we find that instability related to output logit divergence occurs in models with no weight decay regardless of scale, and z-loss resolves this instability. Weight decay also mitigates this instability for the larger models we test.

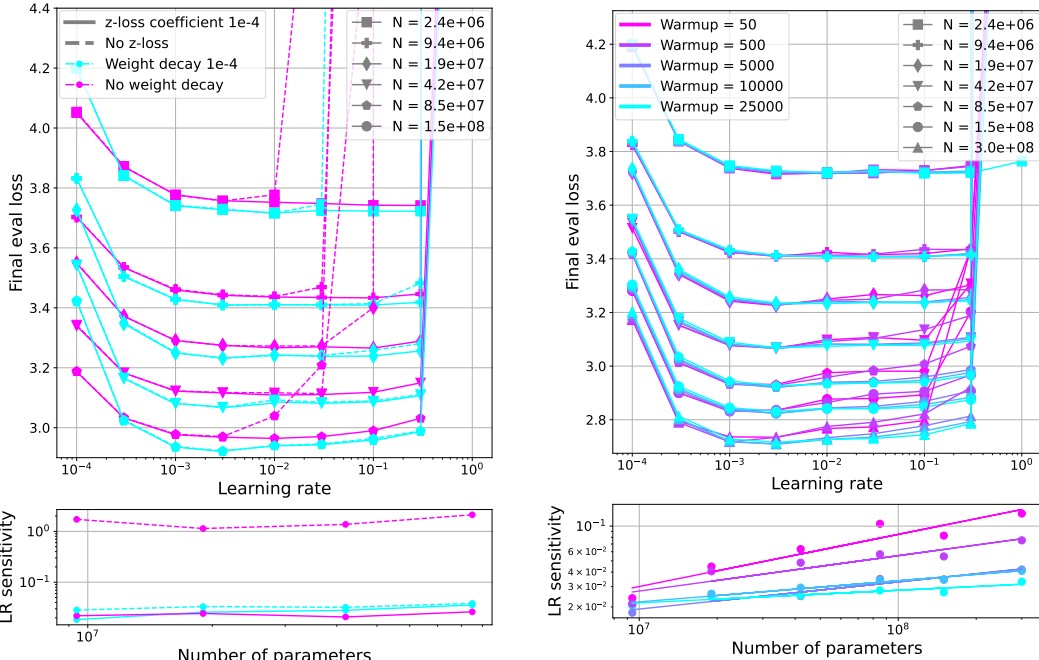

Figure 2: The effect of the output logit divergence instability (Chowdhery et al., 2022) and the z-loss mitigation (Chowdhery et al., 2022) (Section 3.1.2). Models in this experiment have qk-layernorm (Dehghani et al., 2023).

Figure 3: The effect of warm-up length for different model sizes. Longer warm-up reduces LR sensitivity and loss, especially for the larger models we test. Models in this experiment use qk-layernorm (Dehghani et al., 2023).

## 3.2 MEASURING THE EFFECT OF OTHER KNOWN INTERVENTIONS

The previous section used the relationship between learning rate and loss as a useful tool for examining two known instabilities and their mitigation. This raises the question of how other known model and optimizer interventions affect the shape of LR vs. loss curves across scales. In particular, can LR sensitivity help identify additional issues or resolutions when scaling? This section aims to answer this question for common techniques such as warm-up, weight decay, and μParam (Yang et al., 2022). Additional interventions and hyperparamters are tested in Appendix Section D.

### 3.2.1 WARM-UP

As illustrated by Figure 3, a longer warm-up period reduces LR sensitivity. This is most clear for the larger models, which are not stable at LR 3e-1 without long warm-up. The number of total steps is fixed to 1e5 in this experiment, and all models use qk-layernorm. The importance of warm-up for stability has previously been highlighted (Gilmer et al., 2021; Shazeer & Stern, 2018; Liu et al., 2019), although these works do not measure scaling behavior.

### 3.2.2 INDEPENDENT WEIGHT DECAY

Parameterizing weight decay independently of learning rate reduces LR sensitivity, as illustrated in Figure 4. While this was recommended by Loshchilov & Hutter (2019), it is not common practice in the default AdamW implementations of PyTorch (Paszke et al., 2019) or Optax (Babuschkin et al., 2020). We explain the differences below.

For parameters $\theta$, let $\Delta = v/\left(\sqrt{u} + \epsilon\right)$ denote the AdamW update without learning rate or weight decay. For weight decay coefficient $\lambda$, max learning rate $\eta$, and schedule $s_t \in [0, 1]$, Loshchilov & Hutter (2019) recommend the update $\theta \leftarrow \theta - s_t(\eta\Delta - \lambda\theta)$, which we refer to as *independent decay*. On the other hand, the default implementation in PyTorch or Optax applies the update $\theta \leftarrow \theta - s_t\eta(\Delta - \lambda\theta)$, i.e., $\eta$ now scales both terms.

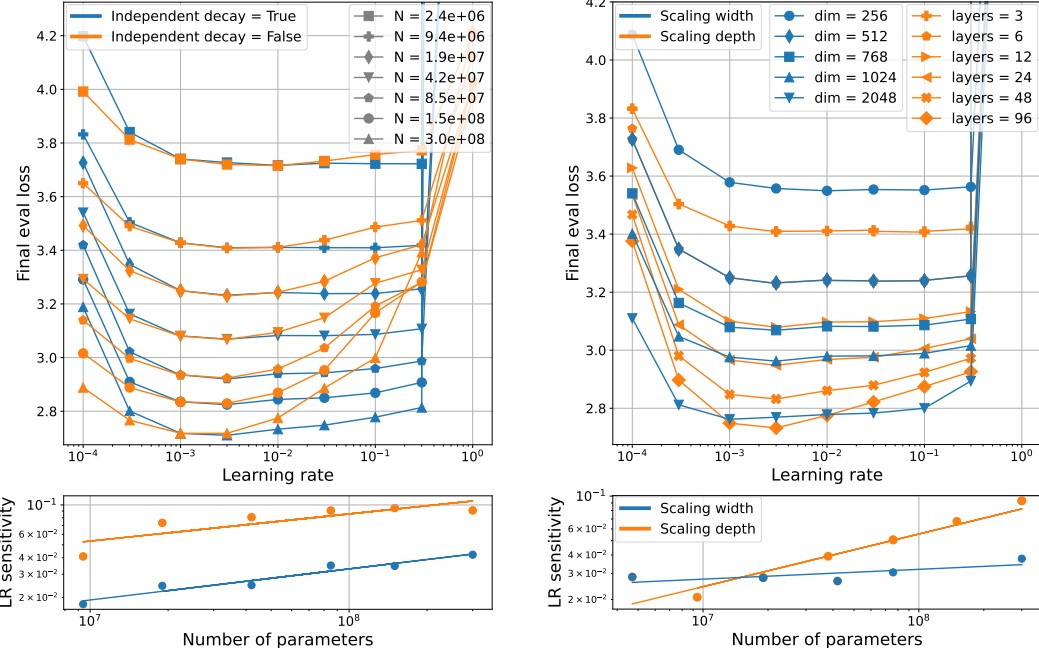

Figure 4: Independently scaling LR without also scaling weight decay reduces LR sensitivity. While this was recommended by Loshchilov & Hutter (2019), it is not common practice in the default AdamW implementations in popular libraries. Refer to Section 3.2.2 for more detail.

Figure 5: Independently scaling depth increases LR sensitivity at a faster rate than scaling width, though also produces a model with lower loss at the largest scale we test. Refer to Appendix Figure G.8 for this experiment without qk-layernorm.

When reporting LR sensitivity without independent decay in Figure 4, we report the minimum LR sensitivity over ranges [1e-4, 1e-1] and [3e-4, 3e-1] because the former is sometimes better centered on the minimum. The default setting in this paper is to use independent decay. When using independent decay we set $\lambda$=1e-4, and without independent decay we set $\lambda$=0.1. A sweep on weight decay values is conducted in Figure G.16.

### 3.2.3 SCALING WIDTH VS. DEPTH

We have so far consistently observed that increasing the number of parameters increases LR sensitivity. We now examine which part of scaling is most responsible.

Our results, illustrated by Figure 5, indicate that scaling depth increases LR sensitivity at a faster rate than scaling width. However, at the largest scale we test, independently scaling depth produces a model with lower validation loss. A validation loss comparison between width scaling, depth scaling, and joint scaling is in Appendix Figure G.9. The standard practice of joint scaling performs best at the largest scale and also has a more reliable scaling prediction when extrapolating.

When scaling depth, we use $d = 512$, and when scaling width, we use 6 layers. The number of heads is scaled proportionally with width, so that the head dimension remains the same.

Figure G.8 repeats this experiment without qk-layernorm, finding that the attention logit growth instability occurs more frequently at scale regardless of whether width or depth are scaled.

### 3.2.4 $\mu$PARAM

Yang & Hu (2021) introduced the $\mu$Param method for parameterizing a neural network. As a product, the optimal LR remains consistent when scaling model width (Yang et al., 2022). This section tests the effect of $\mu$Param on LR sensitivity, and examines whether $\mu$Param alleviates the need for qk-layernorm (Dehghani et al., 2023).

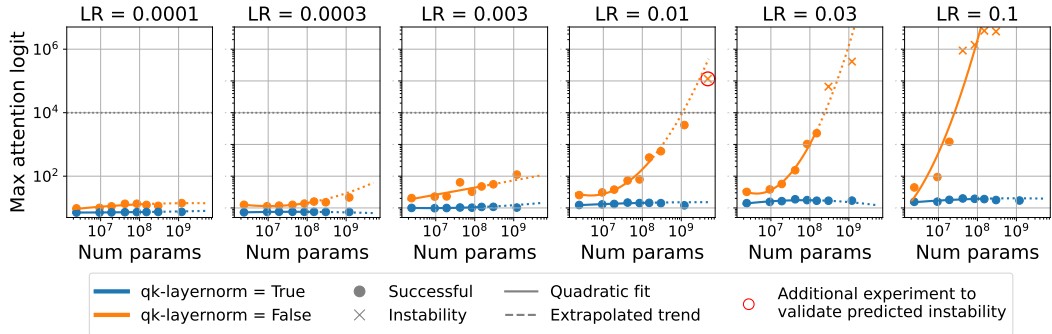

Figure 6: Predicting the attention logit growth instability via scaling behavior of model characteristics. We extrapolate to predict that a larger model will become unstable at LR 1e-2, and run an experiment to confirm the prediction. Refer to Section 3.3 for more information.

As illustrated by Figure G.3, $\mu$Param does succeed in stabilizing the optimal LR at the scale we test. However, $\mu$Param does not improve loss or reduce LR sensitivity in our experiments. Appendix Figure G.4 repeats this experiment without qk-layernorm. Our results indicate that $\mu$Param does not alleviate the need for this intervention at high learning rates. We note that from a practical perspective, reducing LR sensitivity is not important if the optimal LR does not change.

We refer to the variant of $\mu$Param that we use in these experiments as $\mu$Param (simple) because it maintains only the core feature of $\mu$Param. We add additional features from Yang et al. (2022) in Figure G.11 without measurable improvement at the largest scale we test. For $\mu$Param (simple) we make the following changes from our standard baseline: scale the LR for linear layers by base-fan-in/fan-in. For $\mu$Param (full) there are three additional changes: (i) initialize the head with standard deviation $\sqrt{\text{base-fan-in}}/\text{fan-in}$; (ii) change the $1/\sqrt{d_h}$ scaling factor in attention layers to $1/d_h$ where $d_h$ is the head dimension; and (iii) initialize the query projection weights with zeros. For base-fan-in we use the fan-in values for the smallest model we test, which has width 256. We ablate on change (ii) in isolation in Figure G.12. In initial experiments (iii) had no noticeable effect.

### 3.3 PREDICTING ATTENTION LOGIT GROWTH INSTABILITY FROM SCALING BEHAVIOR OF MODEL CHARACTERISTICS

A central question when studying instabilities is whether they can be predicted using small-scale proxy experiments. We now examine whether it is possible to predict the logit growth instability before it occurs using previous runs with smaller models. We track the attention logit maximums across model scales and fit a curve to the data. We use this to predict that a 4.8B parameter model will be unstable at LR 1e-2 without qk-layernorm and run an experiment to confirm this prediction.

Figure 6 plots the number of parameters vs. max attention logit at different learning rate values.[2] At each LR, we fit a quadratic to predict how the max attention logit will change with model scale.

We first noticed that all points with attention logits above 1e4 diverged. Moreover, the quadratic fit predicted that for LR 1e-2 the next model scale would also cross that value. Based on this prediction, we trained a new 4.8B parameter model at LR 1e-2. This model diverged as predicted. Not only do we predict the divergence, but our fit closely extrapolates to predict the value of max attention logit.

Refer to Appendix E for a preliminary experiment on whether we could have predicted that instability arises when the max attention logit exceeds 1e4 without manipulating LR and model size.

### 3.4 SEARCHING FOR NEW INSTABILITIES VIA SCALING TRENDS OF MODEL CHARACTERISTICS

This section examines whether the scaling behavior of model characteristics can be used to predict new issues with the default model and hyperparameter settings.

---

[2]We use block 0, which typically has the largest logits, and consider the value at step 2e3. Much earlier than 2e3 was uninformative, and much later the unstable points had long past diverged.

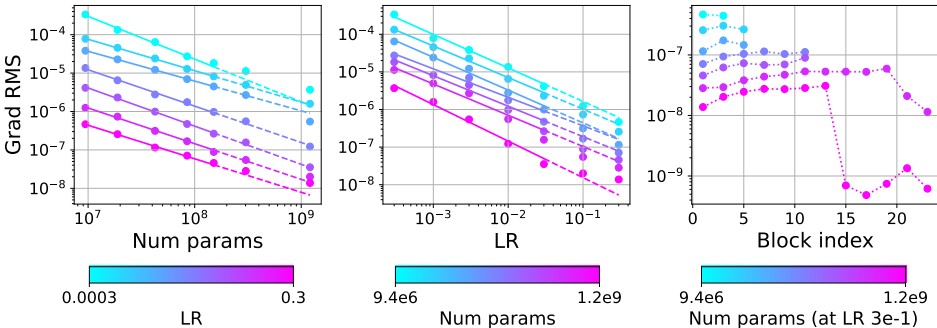

Figure 7: Predicting a potential instability from the scaling behavior of model characteristics. The gradient root mean square (RMS) decreases with num params (left) and learning rate (middle). These trends indicate that hyperparameter adjustment may be required to successfully scale further, as the RMS is approaching the default AdamW $\epsilon$ hyperparameter. If the gradient RMS becomes too small without adjusting $\epsilon$ or weight decay, a layer may collapse. The gradient RMS in the left and middle plot is reported for the first MLP layer of block 0, but we observe similar trends for other layers (e.g., Appendix Figure G.18). Gradient RMS across different blocks is also reported (right). Gradient and update RMS are averaged over the final 500 steps, refer to Appendix Figure G.19 for the data during training.

In Figure 7 we examine scaling trends for the gradient root mean square $\text{RMS}(g) = \sqrt{\mathbb{E}_i\left[g_i^2\right]}$. This figure reports the RMS for the first layer of the MLP, though we observe similar trends for other layers (Appendix Figure G.18).

As models get larger, the value that grad RMS approaches is cause for concern. At the largest scale and learning rate we test, grad RMS is around the default AdamW $\epsilon$ hyperparameter. Recall that the unscaled AdamW update is $\Delta = v/\left(\sqrt{u} + \epsilon\right)$, where $v$ and $u$ are the first and second gradient moment EMA, respectively. If the grad RMS is on the same order as $\epsilon$, then $\Delta$ will decrease in magnitude as illustrated by Figure G.6, and parameters will not receive learning signals as intended.

An obvious mitigation for this issue is to simply lower the AdamW $\epsilon$ hyperparameter from its default of 1e-8. We conduct this experiment for a 4.8B parameter model at LR 0.3 and present the results in Figure G.5. Decreasing $\epsilon$ to 1e-15 improves loss and mitigates a collapse in grad RMS. We believe this improvement will only increase at scale. On the other hand, increasing $\epsilon$ to 1e-6 results in an instability (shown in Figure G.10).

Figure G.6 expands on this result by illustrating the grad and update RMS throughout training at the largest scale and learning rate we test. When the grad RMS reaches $\epsilon$, the update RMS becomes small. Figure G.19 presents data from an analogous experiment at many different scales and LRs, demonstrating that this issue is most apparent for the larger models and LRs we test.

Although we identified the instability above by empirically measuring the scaling behavior of the gradients, a mechanistic explanation exists. As learning rate increases, so does the parameter RMS. A larger parameter RMS leads to a larger RMS for the features output by each Transformer block. Then, the overall output RMS in turn increases with depth due to residual connections. The overall effect is that for larger networks and learning rates, the Transformer output RMS entering the final layernorm will grow. Since the layernorm gradients are scaled by the inverse of their input RMS, the gradient received by the Transformer will shrink. Refer to Appendix C for a detailed discussion.

## 4    RELATED WORK

This paper mainly focuses on the effect of known interventions and instabilities, and so related work has been primarily discussed when relevant. This includes the attention growth instability observed by Dehghani et al. (2023); Zhai et al. (2023a), and the final logit divergence issue encountered by Chowdhery et al. (2022). However, we highlight similar experimental methods in previous work. For instance, Yang et al. (2022) also measure the relationship between LR and loss across scales, but their focus is on centering the optimum (see Section 3.2.4). In addition, Zhai et al. (2023a) elicit instability in base models by doubling learning rate, and Dettmers et al. (2022) measure the presence of outlier features as a function of scale.

There are also important instabilities and related topics we have not directly discussed so far. For instance, we have primarily focused on instabilities that lead to a slow divergence, and we now summarize research on *fast loss spikes*. This instability is characterized by a quick increase in the loss that often eventually recovers.

**The Edge of Stability and fast spikes.** The conventional understanding of gradient descent predicts that loss instability only occurs when the learning rate exceeds $2/\lambda_{\max}(H)$, where $H$ is the Hessian. However recent investigations into large batch neural network training dynamics have revealed a more complicated picture via edge of stability (EoS) (Cohen et al., 2021). When training neural networks with large batch SGD, the loss curvature constantly evolves via the interaction of two processes: progressive sharpening and self stabilization. Progressive sharpening is the empirical observation that when LR $< 2/\lambda_{\max}(H)$, the curvature gradually increases until the stability threshold is violated. When the learning rate becomes too large relative to the curvature, *fast loss spikes* occur and the parameters oscillate into a region with smaller $\lambda_{\max}(H)$ where stable training and progressive sharpening resumes. The latter process where instability results in smaller $\lambda_{\max}(H)$ is self-stabilization, a theoretical model of which is given in Damian et al. (2022). Gradually shrinking $\lambda_{\max}(H)$ via self stabilization was shown to be a primary mechanism behind the success of learning rate warmup in Gilmer et al. (2021), who closely studied the connections between curvature, initialization, architecture and max trainable learning rates.

Cohen et al. (2022) further analyze edge of stability of dynamics with adaptive optimizers, showing that progressive sharpening interacts with both the self-stabilization process and the adaptive optimizer state. This interaction results in the preconditioned sharpness $\lambda_{\max}(P^{-1}H)$ oscillating around an optimizer specific threshold (38/LR in the case of Adam with $\beta_1$=0.9). Adaptive EoS (AEoS) can also result in periodic loss spikes when progressive sharpening pushes the preconditioned sharpness above the stability threshold, however the optimizer hyperparameters play a role. In particular, when LR$>38/\lambda_{\max}(P^{-1}H)$, two mechanisms are now in play to resolve the step size being too big—either $H$ can shrink or $P^{-1}$ can shrink (or both). Cohen et al. (2022) found that when $\beta_2$ is large, $H$ tends to shrink and fast loss spikes result during the process, resembling the self stabilization process observed with gradient descent. However when $\beta_2$ is small, $P^{-1}$ tends to shrink, no loss spikes are observed, and $\lambda_{\max}(H)$ tends to gradually increase throughout training.

It is noteworthy that the adaptive edge of stability process (and the role of $\beta_2$) studied in Cohen et al. (2022) offers a more complete understanding for loss spikes studied in a body of literature (Shazeer & Stern, 2018; Chowdhery et al., 2022; Molybog et al., 2023; Wortsman et al., 2023a; Zhai et al., 2023b; Chen et al., 2021). For example, Shazeer & Stern (2018) argue that during training of Transformers with adaptive optimizers the optimizer update can become too big resulting in a loss spike followed by recovery. This is sometimes attributed to the adaptive optimizer state becoming "stale", which is consistent with the observation the reducing $\beta_2$ resolves the loss spikes (Shazeer & Stern, 2018; Wortsman et al., 2023a; Zhai et al., 2023b). This is perhaps the same observation as Cohen et al. (2022) that reducing $\beta_2$ allows $P^{-1}$ to change quicker to adjust to the process of progressive sharpening. AEoS also offers an explanation for the periodic loss spikes observed when training large transformer models (Molybog et al., 2023).

**Parameter-free methods and more parameterizations.** While our work has studied sensitivity to learning rate, there is also research that aims to eliminate the need to specify a learning rate (Ivgi et al., 2023; Defazio & Mishchenko, 2023). Based on their analysis, Ivgi et al. (2023) set the step size for iteration $t$ to the maximum distance from the initialization divided by the root sum of historical gradient squares. Moreover, while our work investigated $\mu$Param, there are additional parameterizations for which it would be interesting to explore LR vs. loss (Dinan et al., 2023; Yaida, 2022; Bordelon & Pehlevan, 2023; Jacot et al., 2018).

## 5 CONCLUSION

This paper demonstrates that useful insights on instability can be gained from small Transformers. Our results indicate that: (1) instabilities previously reported at scale can be reproduced in small-scale proxy models, facilitating their study without access to large resource pools; 2) instabilities previously reported at scale can be predicted before they emerge by extrapolating from experiments with small-scale proxy models; and 3) new instabilities can be found using small-scale proxy models.

ACKNOWLEDGEMENTS

We thank George Dahl for thorough comments and suggestions, and Hugo Larochelle and Rif A. Saurous for helpful discussion. Also, we thank the members of the Google DeepMind PAGI team for their support of this effort, Noah Fiedel, Noah Constant, Aaron Parisi, Alex Rizkowsky, Avi Singh, Azade Nova, Bernd Bohnet, Daniel Freeman, Gamaleldin Elsayed, Hanie Sedghi, Isabelle Simpson, James Harrison, Jiri Hron, Kathleen Kenealy, Kevin Swersky, Kshiteej Mahajan, Laura Culp, Max Bileschi, Merrie Morris, Rosanne Liu, Yundi Qian, Sharad Vikram, Tris Warkentin.

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

## A  ADDITIONAL INFRASTRUCTURE DETAILS

This Section provides more details on the training infrastructure, which is built on Flax (Heek et al., 2023), Jax (Bradbury et al., 2018), and TPUs (Jouppi et al., 2017). To enable larger model training, we shard the model and optimizer states as in FSDP (Ren et al., 2021), then specify these shadings when compiling with JIT. We use Orbax (Gaffney et al., 2023) for checkpointing, and Grain (Google, 2023) for deterministic data loading. When loading data, sequences are packed so that no padding is required—if a sequence is less tokens than the context length hyperparameter, then an end of sequence token is appended, followed by the beginning of a new sequence.

## B  WHEN IS LEARNING RATE SENSITIVITY A USEFUL METRIC

There are cases where LR sensitivity (defined in Section 2.2) is no longer a useful metric. This section details these scenarios and justifies the use of LR sensitivity for the interventions in this paper.

**Interventions which change the meaning of learning rate.** When an intervention changes the meaning of learning rate then comparing LR sensitivity is not useful. A clear example of this would be taking the square root of the LR before passing it to the optimizer, but there are more subtle cases to be cautious of when using LR sensitivity.

In general, we avoid manipulations where the meaning of LR meaningfully changes. In some cases, we have good empirical evidence that the meaning of the learning rate has not changed when intervening. For instance, the LR vs. loss curves are indistinguishable up to some critical learning rate when using qk-layernorm (Figure 1), adding z-loss (Figure 2), or changing warm-up.

In other cases, such as when testing $\mu$Param (Section 3.2.4), we believe that LR sensitivity is useful despite a per-layer modification of LR. This is because the per-layer LR is manipulated linearly, and this modification does not change for different points on the LR vs loss curve.

**Shifting of the optimal LR.** The definition of LR sensitivity in Section 2.2 does not account for the optimal LR shifting when specifying the LR range $[a, b]$. In practice we recommend shifting the three order of magnitude range $[a, b]$ to correspond with this shift. For instance, we shift the range in Section 3.2.2, as discussed in more detail in the section. However, our main experiments (e.g., Figure 1) do not test at a large enough scale to necessitate this shift.

**LR sensitivity is invariant to loss.** Another limitation of the LR sensitivity metric is that it is invariant to the scale of the loss. If the network consistently achieves random performance across learning rates, then LR sensitivity will be zero. We do not offer a solution to this, and instead recommend that LR sensitivity should always be examined in combination with the LR vs. loss curves as we do in this paper. It is meant as a useful summary of the LR vs. loss curves, not as a metric to optimize in isolation.

## C  PARAMETER AND OUTPUT NORM GROWTH

This section discusses the growth of the parameter norm during Transformer training as previously studied by Merrill et al. (2020); Lee (2023), and relates this phenomenon to the attention logit growth and AdamW epsilon instabilities (Sections 3.1.1 and 3.4, respectively).

An observation of Lee (2023) is that, when using an adaptive optimizer, the movement of parameters can be approximated by a random walk. We show parameter root mean square (RMS) throughout training in Figure C.1[3], which appears to follow a predictable trend. This aligns with the aforementioned observation, and is further supported by Figure C.2. Figure C.2 displays parameter RMS as a function model scale and learning rate, averaged over the last 500 training steps. As before, parameter RMS is determined primarily by learning rate. As parameter RMS grows, we would expect output RMS to also grow. This is validated by Figure C.3, which shows that the RMS of the Transformer block output is mainly determined by learning rate, and follows a very similar trend to parameter RMS.

---

[3]We show parameter RMS for the first MLP layer in various blocks, but expect other layers to exhibit similar trends.

There are two interesting takeaways. First, this observation helps to explain why the attention output logits become large at high learning rates as observed by Dehghani et al. (2023) and Section 3.1.1. This is the only feature in the network we test whose magnitude depends quadratically on parameter RMS. For inputs $X$ with unit RMS, a typical matrix multiply $XW$ with parameters $W$ will result in features $Y$ where RMS$(Y)$ is a linear function of RMS$(W)$. On the other hand, the attention logit entries are computed via $\langle XW_1, XW_2 \rangle$ so depend quadratically on RMS$(W)$. They are therefore the first to become large when the parameter norm grows. Next, this helps to explain the decreasing trend in gradient scale observed in Section 3.4 (Figure 7). In a pre-normalization (Radford et al., 2019) Transformer (Vaswani et al., 2017) there is an output layernorm layer (Ba et al., 2016) after the last Transformer block and before the final linear layer. The gradient from this output layernorm layer is scaled by the reciprocal of the input RMS. In addition to growing with LR, this RMS is growing with depth because of the residual connections (Figure C.3). As the RMS leaving the last Transformer block grows, the gradient received shrinks.

For completeness we now compute the layernorm gradient to input $x$. We assume the input as mean zero and the layernorm has no bias for simplicity. Let

$$z = \text{LayerNorm(x)} = \alpha \cdot \frac{x}{\sqrt{\mathbb{E}_i\left[x_i^2\right] + \epsilon}} = \alpha \cdot \frac{x}{m^{1/2}} \tag{1}$$

where $m = \mathbb{E}_i\left[x_i^2\right] + \epsilon$.

Then

$$\frac{\partial \ell}{\partial x_j} = \sum_k \frac{\partial \ell}{\partial z_k} \frac{\partial z_k}{\partial x_j} \tag{2}$$

$$= \frac{\partial \ell}{\partial z_j} \cdot \frac{\alpha_j}{m^{1/2}} + \sum_k \frac{\partial \ell}{\partial z_k} \cdot \left(-\frac{1}{2}\right) \cdot \frac{\alpha_k x_k}{m^{3/2}} \cdot \frac{2}{n} \cdot x_j \tag{3}$$

$$= \frac{1}{m^{1/2}} \left(\alpha_j \frac{\partial \ell}{\partial z_j} - \frac{x_j}{nm^{1/2}} \sum_k \frac{\partial \ell}{\partial z_k} \alpha_k x_k\right) \tag{4}$$

$$= \frac{1}{m^{1/2}} \left(\alpha_j \frac{\partial \ell}{\partial z_j} - \frac{x_j}{nm^{1/2}} \langle \nabla_z, \alpha \cdot x \rangle\right) \tag{5}$$

Equivalently,

$$\nabla_x = \frac{1}{m^{1/2}} \left(\alpha \odot \nabla_z - \frac{\langle \nabla_z, \alpha \odot x \rangle}{nm^{1/2}} \odot x\right). \tag{6}$$

# D  ADDITIONAL MODEL AND OPTIMIZER INTERVENTIONS

This section recreates the plots from Section 3.2 with additional interventions or hyperparameter changes.

- Changing the number of training steps from 1e5 to 5e4 or 2e5 does not meaningfully change LR sensitivity (Figure G.14).
- We try applying qk-layernorm across the whole model dimension instead of individually per-head with shared paramters. As illustrated in Figure G.13, the latter performs better. We use per-head qk-layernorm as the default in all other experiments.
- Increasing the batch size from 256 to 512 or 1024 does not meaningfully change LR sensitivity (Figure G.15, each batch element contains 512 tokens). When increasing batch size we decrease the number of training steps so that the amount of data seen is constant. We believe a similar effect would be observed if instead we held the number of steps constant because changing the number of steps has no impact on LR sensitivity at batch size 256 (Figure G.14).
- The effect of changing the weight decay from 1e-4 is illustrated in Figure G.16. Increasing decay appears to slightly shift the optimal LR right.
- We find that the logit growth instability is not due to the softmax in the self-attention layer, as it still occurs with a pointwise variant of attention (Figure G.17).

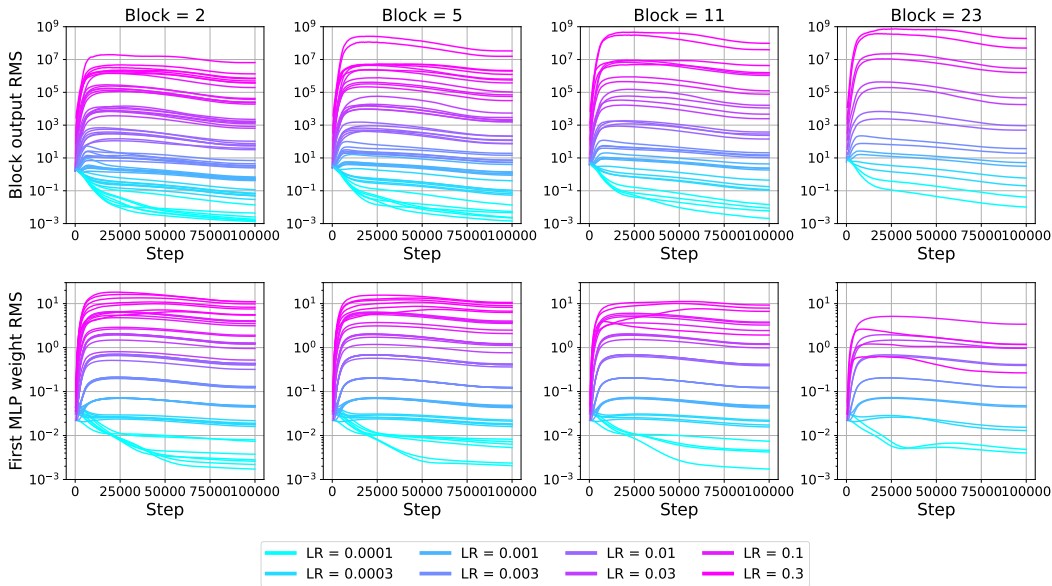

Figure C.1: (Top) The root mean square (RMS) of the Transformer block outputs throughout training. (Bottom) The RMS of the MLP weights throughout training, for the first of the two layers in the MLP. Recall $\text{RMS}(X) = \sqrt{\mathbb{E}_i[X_i^2]}$. RMS is mostly determined by LR, with higher LR corresponding with a higher RMS for block outputs and MLP weights. Different curves at the same learning rate correspond to different model scales. This experiment uses a decoupled weight decay values of 1e-4.

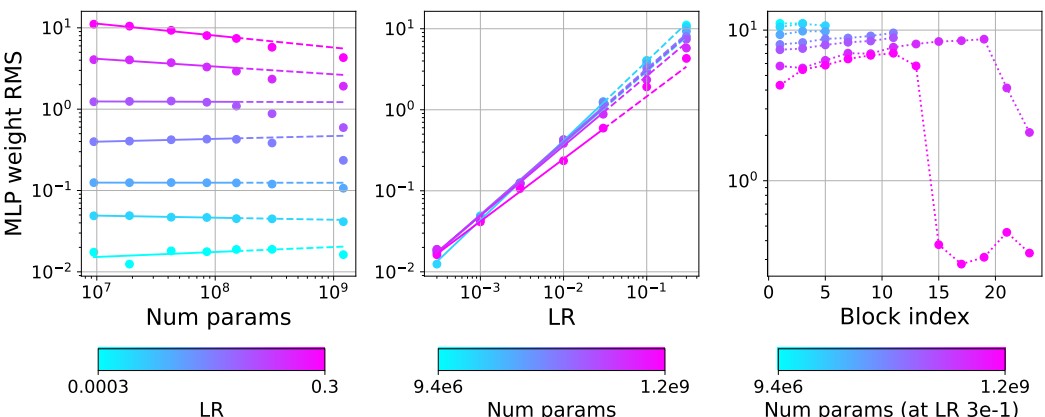

Figure C.2: The root mean square (RMS) of the MLP weights are roughly consistent with scale (left) but increase reliably with learning rate (center). At high learning rates, parameters later in the network can be affected by the AdamW epsilon instability discussed in Section 3.4 (right). This experiment considers the first of the two MLP layers in the block, and data for the first two plots are from block two. RMS is averaged over the final 500 training steps, where $\text{RMS}(X) = \sqrt{\mathbb{E}_i[X_i^2]}$.

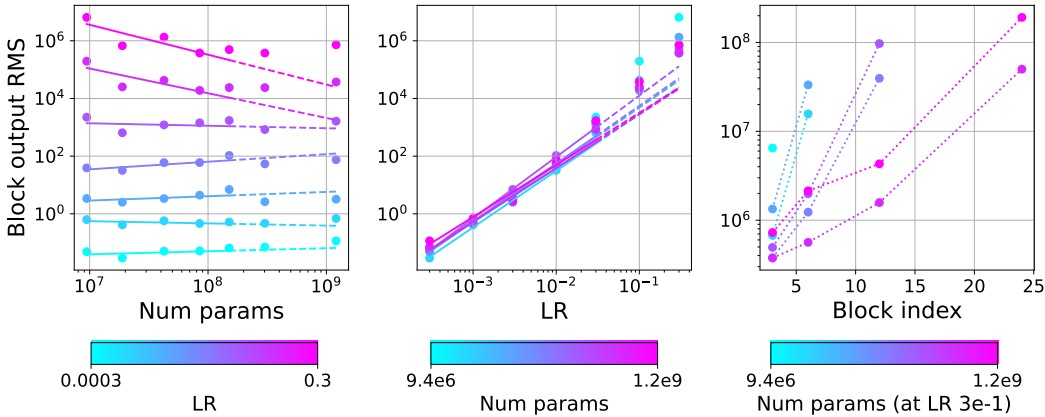

Figure C.3: The root mean square (RMS) of the Transformer block outputs are roughly consistent with scale (left) but increase with learning rate (center). RMS increases deeper in the transformer because of the residual connections, which is shown for very high learning rates (right). The first two plots are for block index two, and RMS is averaged over the final 500 training steps. Recall $\text{RMS}(X) = \sqrt{\mathbb{E}_i[X_i^2]}$.

# E    ADDITIONAL EXPERIMENTS ON PREDICTING THE ATTENTION LOGIT GROWTH INSTABILITY

One question unresolved by our analysis in Section 3.3 is whether we could have predicted that instability arises when the max attention logit exceeds 1e4 without manipulating learning rate and model size. We take initial steps towards an answer by transplanting different values of max attention logit into a small network with 10M parameters. For different constants $\kappa$ we pass the queries and keys through $g(z) = \sqrt{\kappa} \cdot z / \sqrt{\mathbb{E}_i[z_i^2]}$ before computing the attention logits. Results are illustrated in Figure E.1. Loss deteriorates around $\kappa$ =1e3, and by $\kappa$ =1e4 the loss exceeds that of a zero-layer bigram model consisting of the Transformer we use without any self-attention or MLP layers.

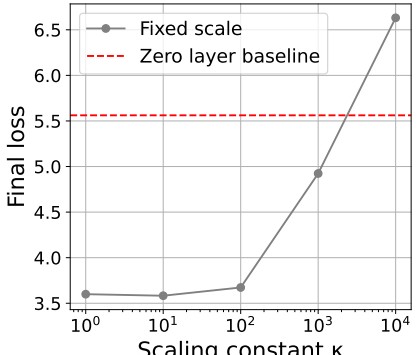

Figure E.1: Enforcing a max attention logit of approximately $\kappa$ in a small model to determine which value of $\kappa$ inhibits learning.

# F    AUTHOR CONTRIBUTIONS

Mitchell Wortsman led the project, ran the experiments and produced the figures, contributed substantially to the infrastructure for experimentation, the framing and direction, and the writing.

Peter J. Liu led the infrastructure and creation of NanoDO for experimentation, provided key insights and advice on multiple technical areas, and contributed to the writing.

Lechao Xiao and Katie Everett contributed to the infrastructure used for experimentation, provided key insight related to parameterization, and contributed to the writing.

Alex Alemi, Ben Adlam, John D. Co-Reyes, Izzeddin Gur, Abhishek Kumar, Roman Novak, Jeffrey Pennington, Jascha Sohl-dickstein, and Kelvin Xu were active participants in weekly brainstorming meetings which motivated, influenced, and elucidated technical concepts pertaining to this work.

Jaehoon Lee and Justin Gilmer were senior authors advising on the project, contributed substantially to the framing and direction, provided key insight and advice on multiple technical areas, and contributed to the writing. Jaehoon led the connection with output norm growth. Justin proposed to plot loss as a function of learning rate for different model sizes, and performed initial experiments demonstrating that attention logit growth could be reproduced at high learning rates in small models.

Simon Kornblith was the lead advisor on the project, contributing substantially to the framing, direction, infrastructure, and writing. Simon initially brainstormed the project with Mitchell, and was Mitchell's host for the summer internship during which this research was conducted, providing substantial technical support.

# G    ADDITIONAL FIGURES

This Section contains the additional Figures referenced in the main text and appendix.

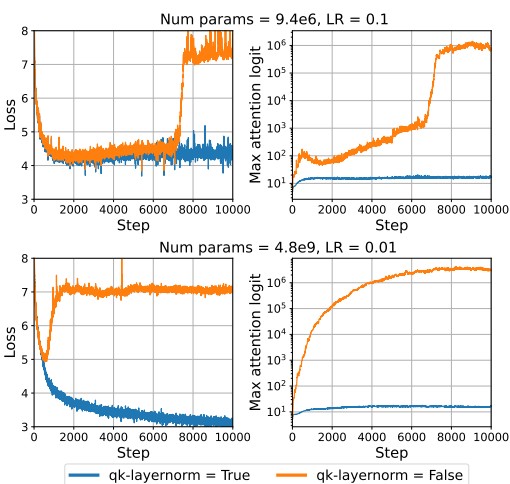

Figure G.1: The attention logit growth instability (Dehghani et al., 2023; Zhai et al., 2023a) appears in small models at high learning rates. The mitigation of applying qk-layernorm proposed by Dehghani et al. (2023) is equally effective in the small-scale regime. The max attention logit is reported for layer 0, which we typically observe to have the largest logit values.

Figure G.2: An example of the output logit divergence instability (Chowdhery et al., 2022) (Section 3.1.2) in a 2.4M parameter Transformer at learning rate 0.1.

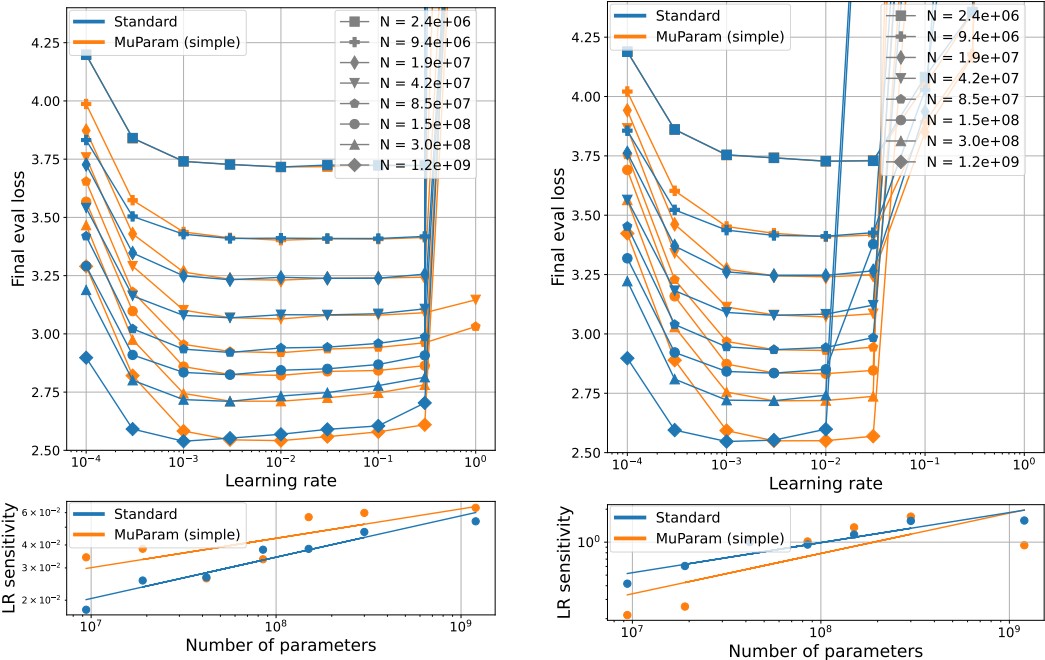

Figure G.3: Measuring the effect of $\mu$Param on LR sensitivity for models with qk-layernorm (Dehghani et al., 2023). $\mu$Param succeeds in stabilizing the optimal LR, though it does not improve loss or reduce LR sensitivity. For more information refer to Section 3.2.4.

Figure G.4: The effect of $\mu$Param on LR sensitivity for models without qk-layernorm (Dehghani et al., 2023). $\mu$Param succeeds in stabilizing the optimal LR, but does not alleviate the need for qk-layernorm. For more information refer to Section 3.2.4.

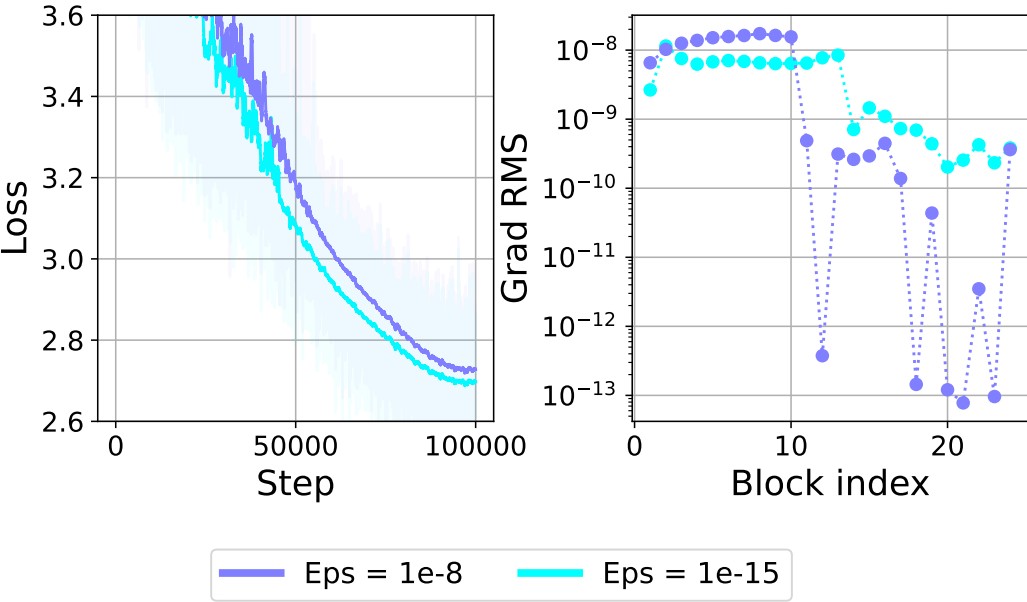

Figure G.5: Decreasing the AdamW $\epsilon$ from its default value of 1e-8 to 1e-15 improves loss for a 4.8B parameter model at LR 0.3. When increasing $\epsilon$ to 1e-6, loss diverged. Grad RMS is averaged over the final 500 steps for the first layer in the MLP; refer to Figure G.6 for data throughout training.

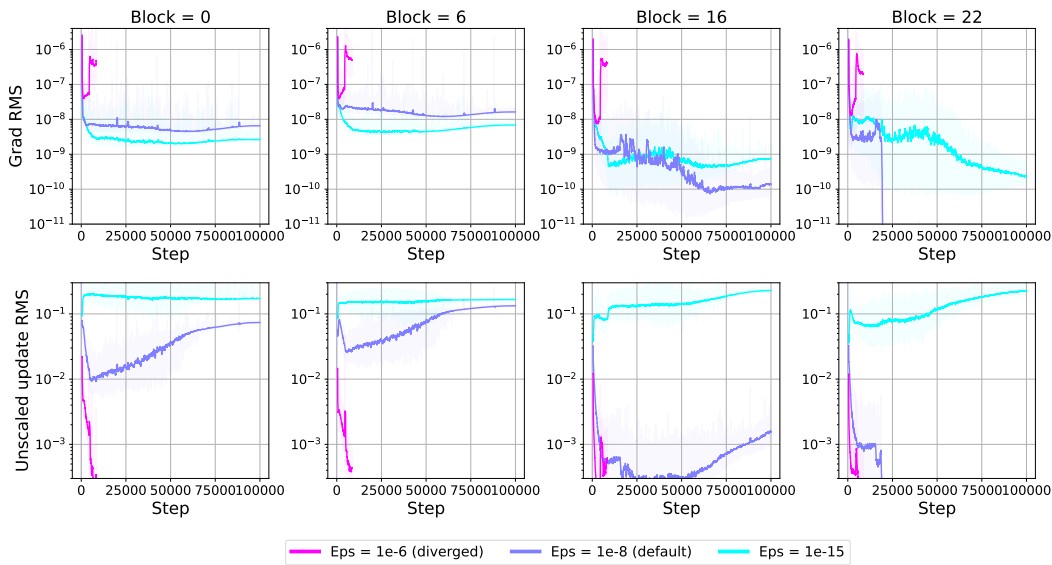

Figure G.6: The top row displays the root mean square (RMS) of the gradient for the first MLP layer at different blocks throughout the network. When the grad RMS drops below the AdamW $\epsilon$ hyperparameter, the magnitude of the update decreases, as illustrated by the bottom row. Experiment conducted with a 4.8B parameter model trained with LR 0.3. The experiment with $\epsilon$ = 1e-6 was stopped when loss diverged.

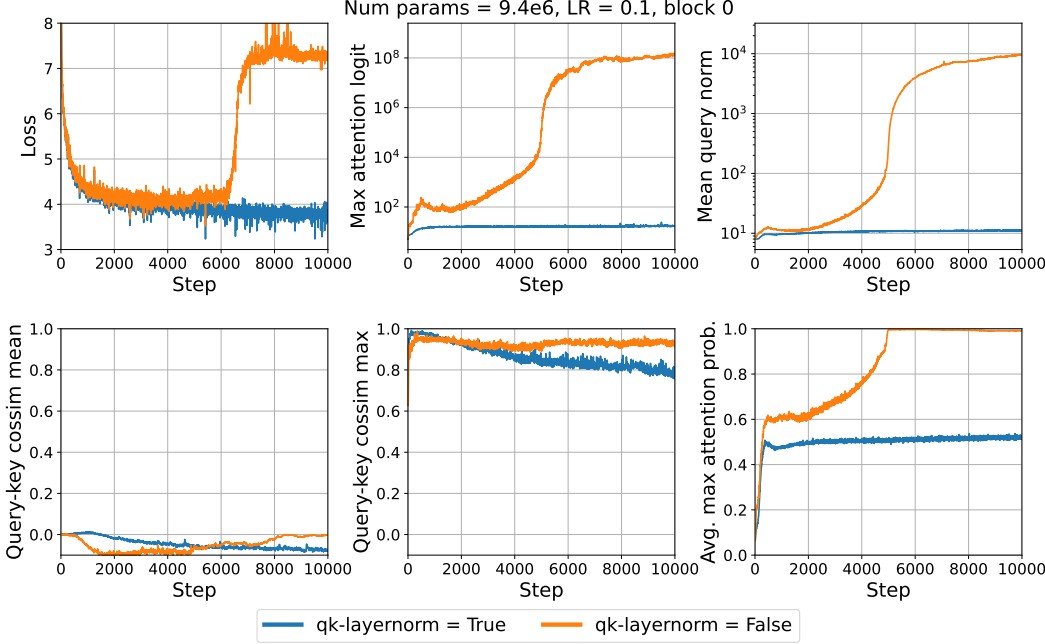

Figure G.7: The logit growth instability (Dehghani et al., 2023; Zhai et al., 2023a) occurs when the norm of the query and keys increases, not due to an increase in their cosine similarity.

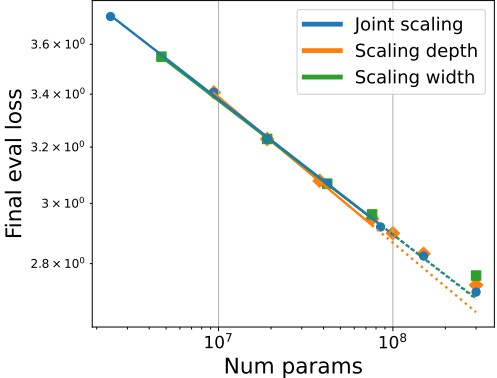

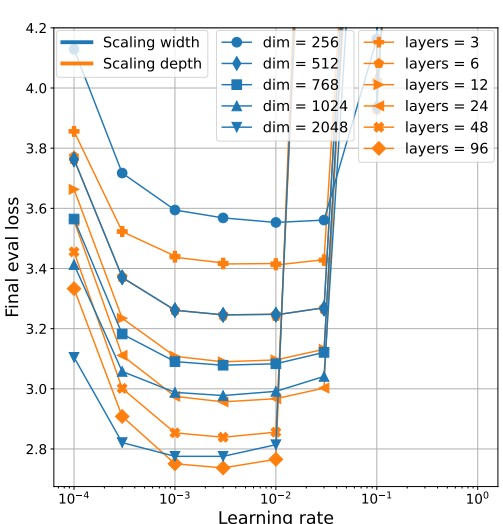

Figure G.8: The effect of scaling width vs. scaling depth without qk-layernorm (Dehghani et al., 2023).

Figure G.9: Jointly scaling width and depth leads to lower loss than independently scaling depth or width at the largest scale we test. It also leads to a more reliable scaling prediction when extrapolating from models with less than 1e8 parameters. Best loss is reported in a sweep over learning rates.

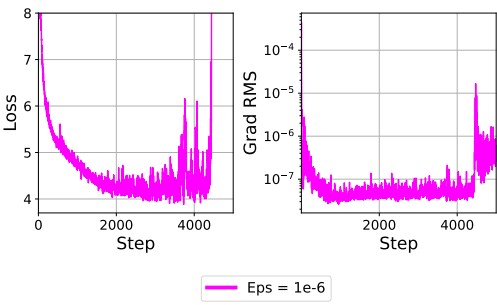

Figure G.10: Increasing the AdamW $\epsilon$ from its default value of 1e-8 to 1e-6 causes a loss divergence for a 4.8B parameter model at LR 0.3. Grad RMS is for the first layer in the MLP.

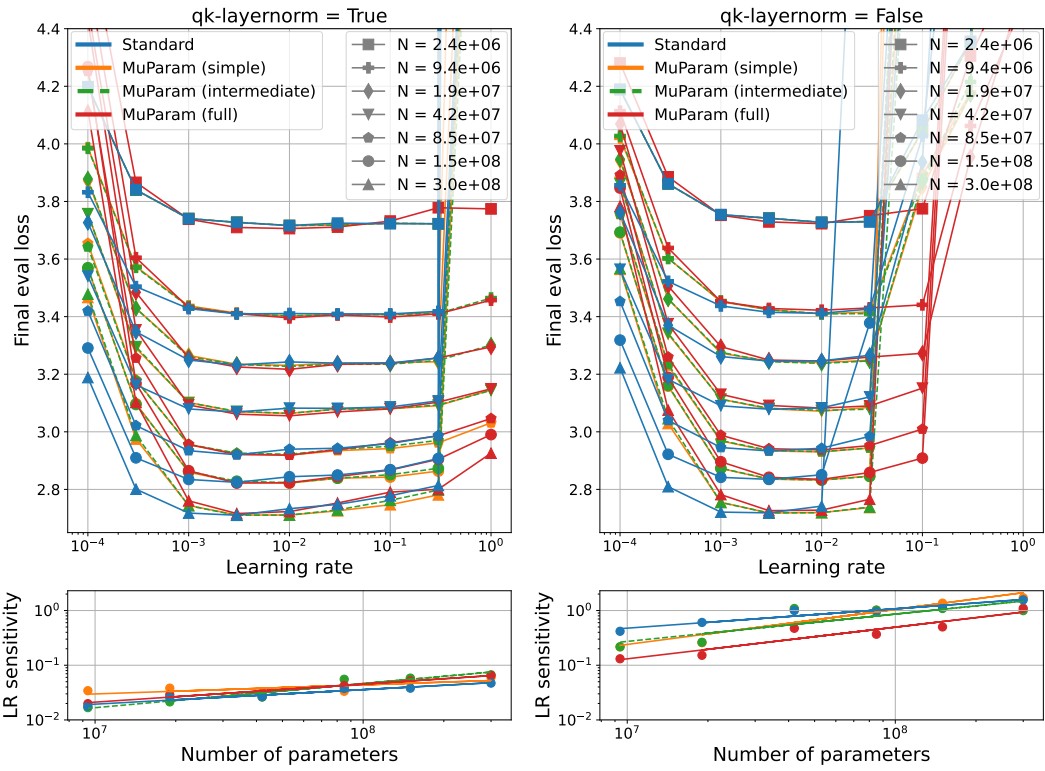

Figure G.11: Comparing $\mu$Param (full), which implements $\mu$Param as described in Yang et al. (2022) with and without qk-layernorm, with $\mu$Param (simple) and $\mu$Param (intermediate). There are four changes in $\mu$Param (full), (i) Scale the LR for linear layers by base-fan-in/fan-in, (ii) initialize the head with standard deviation $\sqrt{\text{base-fan-in}}/$fan-in. (iii) change the $1/\sqrt{d_h}$ scaling factor in attention layers to $1/d_h$ where $d_h$ is the head dimension, and (iv) initialize the query projection weights with zeros. $\mu$Param (intermediate) consists of (i) and (ii), while $\mu$Param (simple) is only (i). With $\mu$Param (full) and qk-layernorm, the model trains without diverging at LR 1. However at the best LR there is no measurable improvement over $\mu$Param (simple) at the largest scale we test.

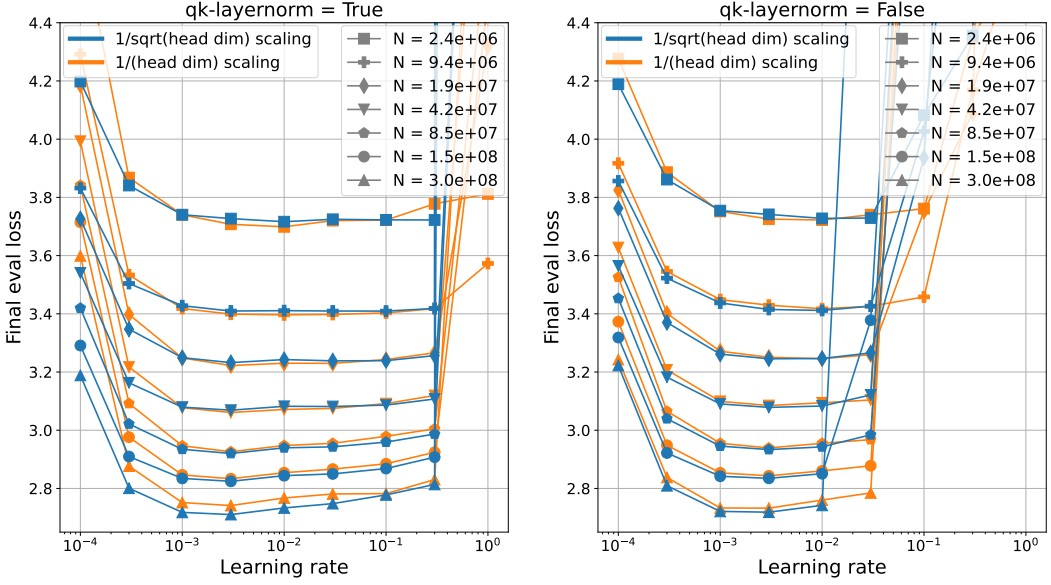

Figure G.12: Measuring the effect of changing the $1/\sqrt{d_h}$ term in attention to $1/d_h$, where $d_h$ is head dimension. Vaswani et al. (2017) use $1/\sqrt{d_h}$ while Yang et al. (2022) use $1/d_h$.

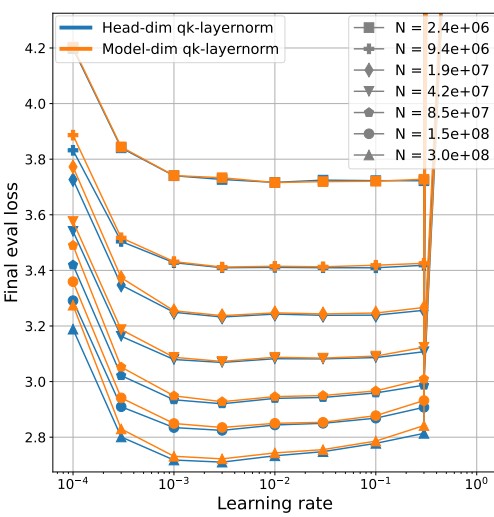

Figure G.13: We achieve slightly better performance when applying qk-layernorm individually per-head instead of across the model dimension. The per-head variant has only head-dim learnable parameters instead of model-dim parameters. We use the per-head variant as the default in this paper, and we never use biases.

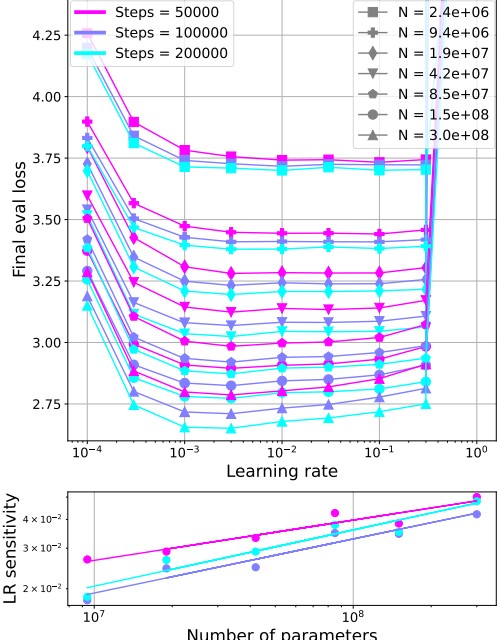

Figure G.14: Changing the number of total training steps from 1e5 to 5e4 or 2e5 does not have a large effect of the shape of the learning rate vs. loss curves at the scales we test.

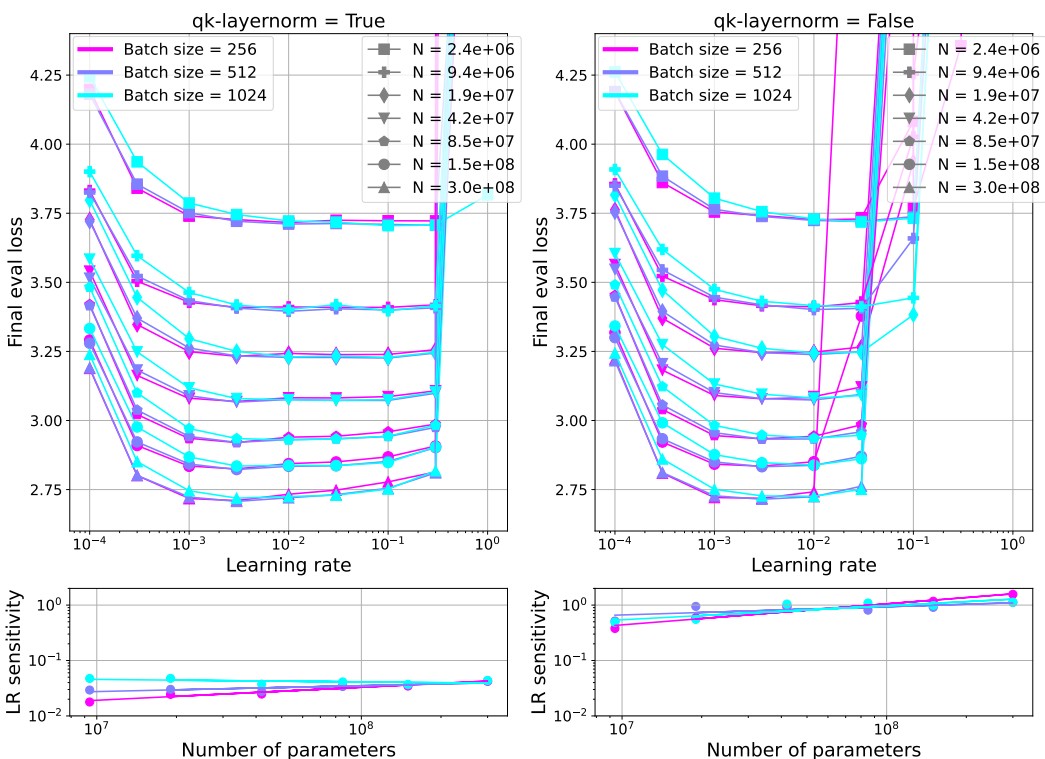

Figure G.15: Increasing the batch size from 256 to 512 or 1024 does not have a large effect on the shape of the learning rate vs. loss curves at the scales we test. Each batch element contains 512 tokens, and we use 256 as the default.

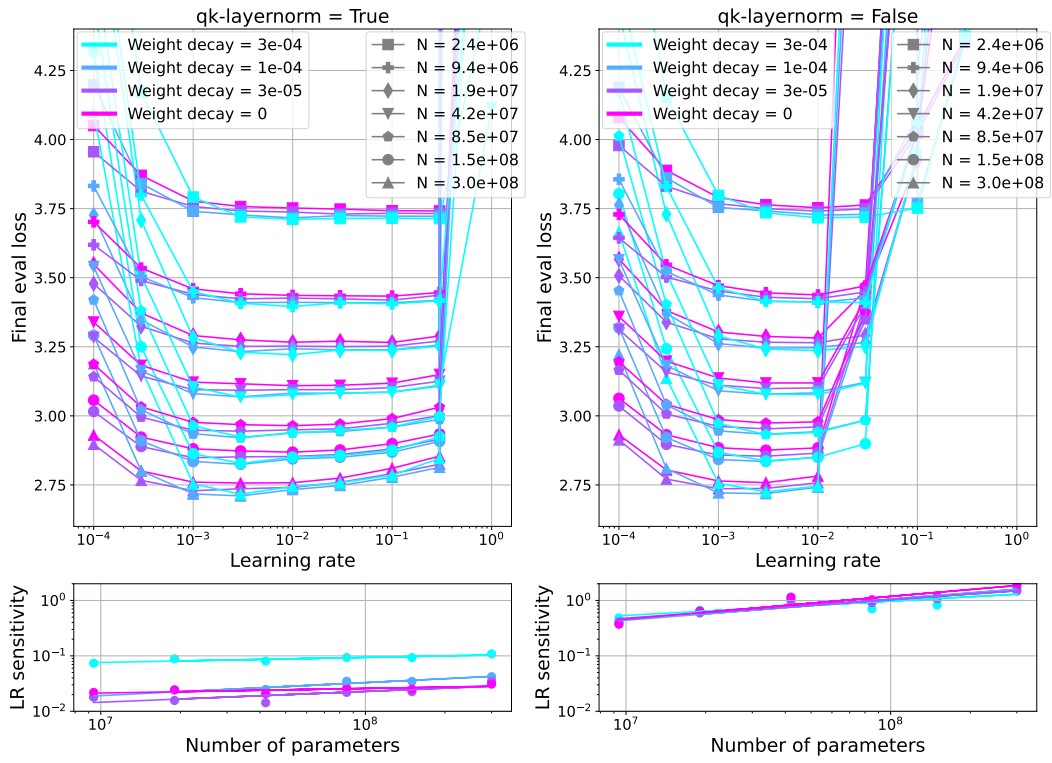

Figure G.16: The effect of weight decay on LR sensitivity. We use independent weight decay as described in Section 3.2.2 and recommended by (Loshchilov & Hutter, 2019).

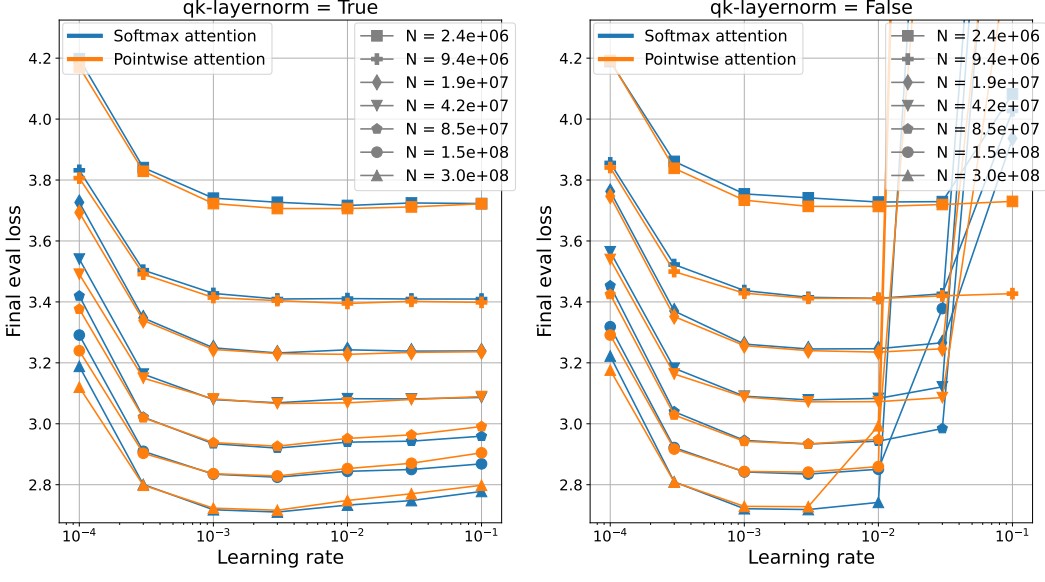

Figure G.17: The logit growth instability occurs even without softmax. For the pointwise variant of attention here, we replace softmax with squared-relu as described by (Hua et al., 2022). As recommended in (Wortsman et al., 2023b) we add a scaling factor which depends on sequence length. In this case, we use inverse square root.

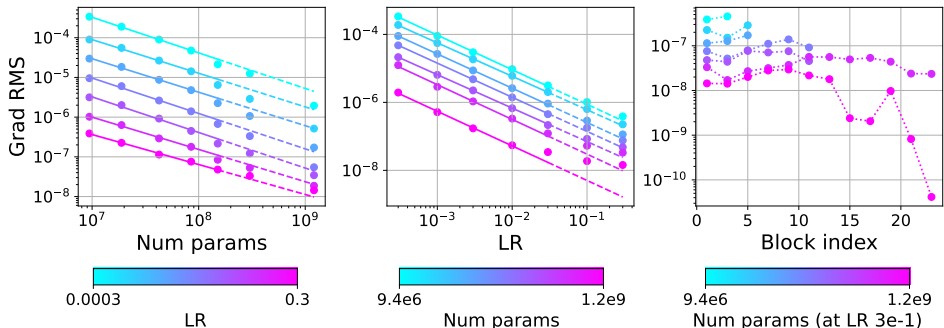

Figure G.18: Recreating Figure 7 with the kernel projection instead of the first MLP layer.

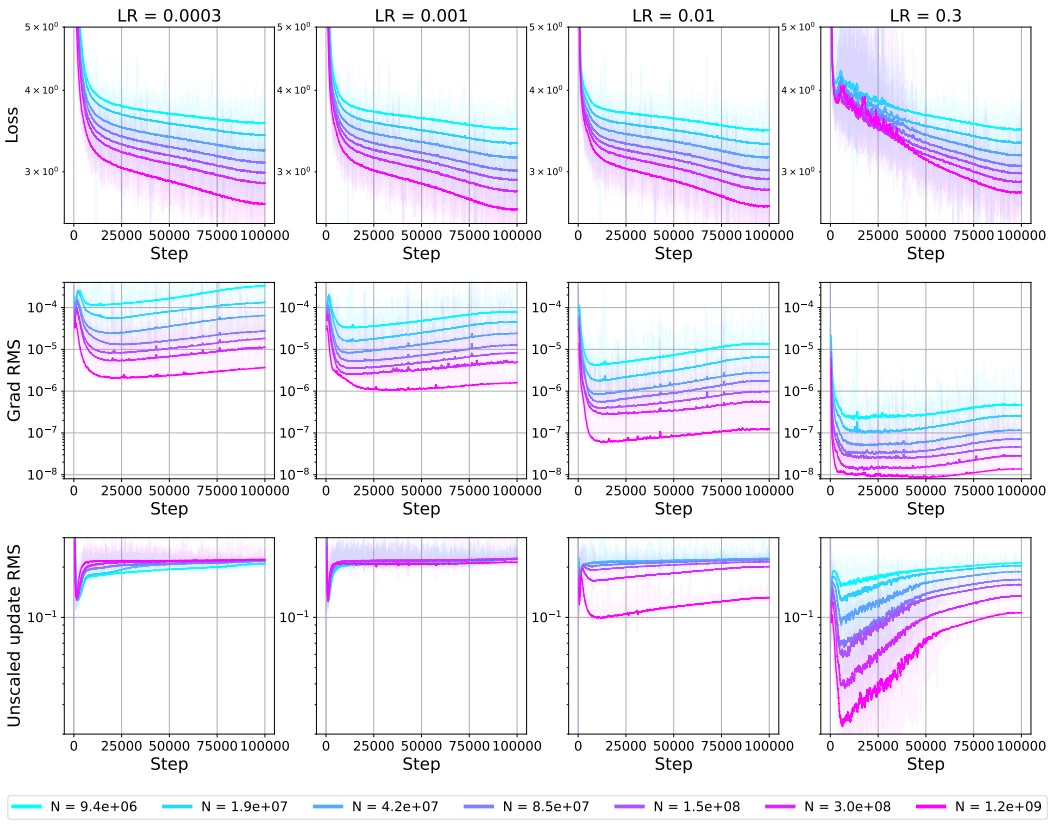

Figure G.19: For various learning rates and model sizes we display the gradient root mean square (RMS), and the unscaled update RMS. The unscaled udpate is the update returned by the optimizer before scaling by learning rate. The gradient and update are shown here for the first MLP layer of the Transformer. The update RMS falls when the grad RMS approaches the AdamW $\epsilon$ of 1e-8.

