# OpenReview forum: "Small-scale proxies for large-scale Transformer training instabilities"
_ICLR.cc/2024/Conference — ICLR 2024 oral_

### Official Review · Reviewer_hm5n · 2023-10-18

**Soundness:** 4 excellent
**Presentation:** 4 excellent
**Contribution:** 3 good
**Rating:** 8
**Confidence:** 4

**Summary:**

The paper studies instabilities of transformers on a smaller scale. Specifically, the authors performs ablation experiments over learning rates of small transformers, and finds that techniques that are known to improve stability for large transformers also improve the stability of small transformers when using high learning rate. Among other things, the authors show 1) that qk normalization enables higher LR, 2) that the z-loss enables higher LR 3) LR warmup makes the model less LR sensitive, 4) Independently parametrizing WD and LR makes the model less LR sensitive, 5) model LR sensitivity grows faster with depth than width.

**Strengths:**

Large scale transformers are expensive, important and suffer from instabilities. Providing a small-scale proxy model is impactful.

The paper is well written and the experiments are cleanly described.

The observations on independent weight decay and the scaling of the gradient RMS are relatively novel.

**Weaknesses:**

A significant part of the paper is dedicated to replicating observations made in large transformers to small transformers. The utility of this is a little unclear. While it demonstrates that a small model with high LR could serve as a proxy for a larger model, it doesn’t demonstrate any new insights regarding large models. It would be more impactful if the authors would make previously unknown observations at a small scale, and then show that they hold at a larger scale.

Section 3.3 reads a little anecdotal to me. A more systematic study would be better.

**Questions:**

Should LR sensitivity be normalized somehow? The optimal loss scales with model size, so the delta in eval loss between models of different scales are not really comparable.

Will the code be open sourced?

---

> ### Author Response · Authors · 2023-11-22
>
> Thank you for your detailed review, we are glad that you found the paper well written and the experiments clearly described. We hope that the following addresses your reported weaknesses and questions:
>
> > A significant part of the paper is dedicated to replicating observations made in large transformers to small transformers. The utility of this is a little unclear. While it demonstrates that a small model with high LR could serve as a proxy for a larger model, it doesn’t demonstrate any new insights regarding large models. It would be more impactful if the authors would make previously unknown observations at a small scale, and then show that they hold at a larger scale.
>
> We believe that our research yields new insights. For instance, the instability described in Section 3.4 has not previously been documented. Moreover, we believe that establishing small-scale models as reliable proxies is a useful new development which opens the door for further research on understanding/mitigating instabilities.. For instance, future research could confirm the hypothesis in Section 3.1.1, which is that large attention logits result from quadratic dependence on parameter norms
>
> > Section 3.3 reads a little anecdotal to me. A more systematic study would be better.
>
> We agree with this and have revised the paper to list this as a limitation in Section 5.
>
> > Should LR sensitivity be normalized somehow? The optimal loss scales with model size, so the delta in eval loss between models of different scales are not really comparable.
>
> Thank you for this interesting suggestion. We usually observe that LR sensitivity increases with scale. If we understand correctly, normalizing would only exaggerate this effect, because it would amplify differences at lower loss values?
>
> > Will the code be open sourced?
>
> We believe that the important parts of the code we use are open source, and draw attention to them here: 1) qk-layernorm is implemented as part of MultiHeadDotProductAttention in Flax https://github.com/google/flax/blob/70214f4ecee59fdd82c07b692d4ad3d47149fe3d/flax/linen/attention.py#L438-L442, 2) z-loss is implemented as part of t5x: https://github.com/google-research/t5x/blob/main/t5x/losses.py#L53-L56.

---

> > ### Comment · Reviewer_hm5n · 2023-11-22
> > **reply**
> >
> > I thank the author for their reply and will retain my score.
> >
> > Regarding the normalization my thought is that loss will saturate with model scale. A small increase in loss for a large model might correspond to the model performing like a model at half its parameter size. For a small model with larger loss, a small increase in loss might just correspond to the model performing like a model with 90% of the parameter count. But taking this into account would probably just make the definition more complicated.

---

### Official Review · Reviewer_Fq7A · 2023-10-26

**Soundness:** 3 good
**Presentation:** 3 good
**Contribution:** 3 good
**Rating:** 8
**Confidence:** 3

**Summary:**

An experimental paper. The authors' main point seems to be about the attention and output logits in transformer yielding instabilities. This may be a valid point, although it is not very how to mitigate this problem, and it is hard to be completely convinced that this is *the* reason for instability of large transformers. That being said, some of the experiments are valuable and help us a little bit to understand some issues that may arise in the training of transformers. There is an emphasis on considering the learning rate size.

The suggested experimental evidence supporting this claim is "val loss vs learning rate" curves. However, (1) there is no surprise in training divergence when lr becomes too large, and (2) I do not see any experimental evidence that divergence is indeed caused by the considered instabilities and not by something else.

The paper also studies how "learning rate sensitivity" is affected by certain design choices. Learning rate sensitivity is defined as the average of "excess val loss" over learning rate range. However, the choice of particularly this metric does not seem well-motivated. Do authors use uniform distribution over lr? If yes, why not uniform over log(lr)? Why not simply use maximal stable lr?

One insight which seems useful is that default eps=1e-8 in AdamW might appear too large and cause updates to vanish.

**Strengths:**

Understanding stability of transformer training is an important problem. The hypothesis that instabilities may be related to attention logits is not without interest. The numerical experiments seem to be very carefully made, and overall they bring some value. I thank the authors for the clarifications.

**Weaknesses:**

The structure of the paper is a little weird (the conclusion is very short and contains no useful information, the discussion of existing results is just put at the end without much being done from it, the main points seem to be made in the figures. ). The way the logits in the attention mechanism pose problem is not made super clear or intuitive (obviously, it's a little hard to prove something, but at least some intuition would be appreciated). For instance, we learn that high enough learning rate will pose problem at some point, but that's the kind of things that is not surprising. Does this validate the whole hypothesis?
Note: the concerns have been addressed.

**Questions:**

Are we really sure that the reason for large transformers not training well is logit divergence? What are other possible problems? What do we learn in the end from your analysis? Is it clear that such problems don't arise in other architectures?

---

> ### Author Response · Authors · 2023-11-22
>
> We thank you for your comprehensive review, and hope that the following addresses the questions and weaknesses:
>
> In response to the questions:
>
> > Are we really sure that the reason for large transformers not training well is [attention and output] logit divergence?
>
> We do not intend to claim that logit divergence is the only reason for large-scale Transformer training instability. However, we are relatively certain that the specific forms of training instability we study in Section 3.1 are caused by logit divergence. There are a few experimental results which support this conclusion:
> Intervening to reduce the attention logit norms via qk-layernorm mitigates instability at high learning rates (Figure 1 and F.1).
> In Figure 6, all runs which have an attention logit maximum above 10^4 are unstable, while all other runs are stable.
> Intervening to increase the attention logits in small (10M parameter) models at low learning rates deteriorates accuracy (Appendix E).
> Intervening to reduce the divergence of the output logits via z-loss mitigates instability at high learning rates (Figure 2 and F.2).
>
> > What are other possible problems?
>
> There are a number of other possible problems with Transformer training. One which we highlight in Section 3.4 is vanishing updates due to the AdamW epsilon hyperparameter. Another potential problem is slower learning due to “fast loss spikes”, as discussed in Section 4. Overall, we believe our results suggest that additional potential instabilities can be studied using small-scale proxy models.
>
> > What do we learn in the end from your analysis?
>
> Overall we learn that we can reproduce, study, and predict Transformer training instabilities at large-scale using small-scale proxy models. The revised conclusion shares some highlights: “This paper demonstrates that useful insights on instability can be gained from small Transformers. Our results indicate that: (1) instabilities previously reported at scale can be reproduced in small-scale proxy models, facilitating their study without access to large resource pools; 2) instabilities previously reported at scale can be predicted before they emerge by extrapolating from experiments with small-scale proxy models; and 3) new instabilities can be found using small-scale proxy models.”.
>
> > Is it clear that such problems don't arise in other architectures?
>
> We believe that further experimental evidence would be required to make concrete conclusions about other architectures. However, a similar instability to attention logit growth would likely not occur in MLPs or CNNs. This is because, as we hypothesize (in Section 3.1.1), this instability may result from the quadratic dependence of attention logits on parameter norms. This quadratic dependence does not occur in MLPs or CNNs.
>
> In response to the weaknesses:
>
> > The structure of the paper is a little weird (the conclusion is very short and contains no useful information, the discussion of existing results is just put at the end without much being done from it, the main points seem to be made in the figures. ).
>
> We acknowledge this oversight. In our revision we have changed the conclusion so that it is more informative. It now reads: “This paper demonstrates that useful insights on instability can be gained from small Transformers. Our results indicate that: (1) instabilities previously reported at scale can be reproduced in small-scale proxy models, facilitating their study without access to large resource pools; 2) instabilities previously reported at scale can be predicted before they emerge by extrapolating from experiments with small-scale proxy models; and 3) new instabilities can be found using small-scale proxy models.”
>
>
> > The way the logits in the attention mechanism pose problem is not made super clear or intuitive (obviously, it's a little hard to prove something, but at least some intuition would be appreciated). For instance, we learn that high enough learning rate will pose problem at some point, but that's the kind of things that is not surprising. Does this validate the whole hypothesis?
>
> We agree that intuition could be helpful. However, further experiments may be required to make concrete conclusions. One reason a large attention logit could cause an issue is by saturating softmax and restricting gradient flow.
>
> In response to additional comments:
>
> > The authors' main point seems to be about the attention and output logits in transformer yielding instabilities. This may be a valid point, although it is not very how to mitigate this problem [...].
>
> We believe our results indicate that z-loss and qk-layernorm are effective mitigations which enable stable Transformer training even at high learning rates.
>
> > Do authors use uniform distribution over lr? If yes, why not uniform over log(lr)?
>
> The learning rate values we use in our experiments are (approximately) evenly spaced on a log scale: [1e-4, 3e-4, 1e-3, 3e-3, 1e-2, 3e-2, 1e-1, 3e-1, 1e0].

---

### Official Review · Reviewer_SBqh · 2023-11-01

**Soundness:** 3 good
**Presentation:** 3 good
**Contribution:** 2 fair
**Rating:** 8
**Confidence:** 2

**Summary:**

This article studies optimization instabilities observed in the training of large transformer-based models. The central contribution is the reproduction and analysis as small scale of instabilities that were previously observed on large-scale models. This allows to study the stability of those models without needing the large computing power required for large-scale training.

Two central kind of instabilities are studied by the authors: the growth of logic in attention layers, and the divergence of output logits of the model. In both cases, it is experimentally shown that those instability can be reproduced on small models when using a large learning rate, and that the mitigation techniques that were developed for large models are equally effective in this context. The core tool used for this analysis is introduced to be the measure of the sensibility of the model performance to the learning rate used for the optimization, and the experimental results show that those mitigations tend to reduce that sensibility, stabilizing the training.

The authors finally extend their analysis to study the impact of several other interventions that have been proposed, such as the parameterization of the trainable weights, the integration of weight decay in the optimizer, the scaling of the model size and the use of warm-up periods.

**Strengths:**

This is an extensive and detailed experimental study of the stability of transformer models with regard to the training learning rate and the various mitigation methods that have been considered.

The experimental setup is described with abundance of details, the conducted experiments are well motivated and presented, and the analysis tools (as the LR sensibility) allows a synthetic and clear summary of the impact of the parameters & methods evaluated.

I believe this article has the potential to provide a wealth of useful information and heuristics for practitioners working with such models.

**Weaknesses:**

While I am not extremely familiar with the large-transformer-models community, I am under the impression that the pool of persons effectively concerned by this work is very small. As the authors note, training such large models is very computationally expensive, and currently only very few groups have the means to train such models.

As a result, I wonder if this subject might be in practice rather niche, in terms of how much of the community could actually use it.

**Questions:**

I don't have more questions.

---

> ### Author Response · Authors · 2023-11-22
>
> We thank you for the thoughtful review, and are glad that you found this paper to have “potential to provide a wealth of useful information”. We hope to resolve your question below:
>
> > While I am not extremely familiar with the large-transformer-models community, I am under the impression that the pool of persons effectively concerned by this work is very small. As the authors note, training such large models is very computationally expensive, and currently only very few groups have the means to train such models. As a result, I wonder if this subject might be in practice rather niche, in terms of how much of the community could actually use it.
>
>
> The expense of training large models indeed means that few researchers will observe the training instabilities we study in practice, but it also means that research aimed toward understanding instability can be immensely practically impactful. Despite limits on the size of models they can train, academic researchers have successfully studied many different aspects of Transformer-based models. There is some published research regarding Transformer training instability [1,2,3,4,5,6,7], but it has been less popular than e.g. architecture, likely due to perceived difficulty of studying training instability on a small compute budget. Our goal is to enable such studies by showing that it is possible to elicit training instabilities representative of those observed at large scale in smaller models. While Transformer training instability is currently a niche topic, we believe that it is both scientifically interesting and practically impactful, and we hope our work is a useful step in enabling more researchers to explore this area.
>
> [1] Gilmer et al., 2021. A Loss Curvature Perspective on Training Instability in Deep Learning. https://arxiv.org/abs/2110.04369.
>
> [2] Cohen et al., 2021. Gradient Descent on Neural Networks Typically Occurs at the Edge of Stability. https://arxiv.org/abs/2103.00065.
>
> [3] Cohen et al., 2022. Adaptive Gradient Methods at the Edge of Stability. https://arxiv.org/abs/2207.14484.
>
> [4] Damian et al., 2022. Self-Stabilization: The Implicit Bias of Gradient Descent at the Edge of Stability. https://arxiv.org/abs/2209.15594.
>
> [5] Molybog et al., 2023. A Theory on Adam Instability in Large-Scale Machine Learning. https://arxiv.org/abs/2304.09871.
>
> [6] Zhai et al., 2023. Stabilizing Transformer Training by Preventing Attention Entropy Collapse. https://arxiv.org/abs/2303.06296.
>
> [7] Dehghani et al., 2023. Scaling Vision Transformers to 22 Billion Parameters. https://arxiv.org/abs/2302.05442.

---

### Official Review · Reviewer_TVMA · 2023-11-03

**Soundness:** 3 good
**Presentation:** 3 good
**Contribution:** 3 good
**Rating:** 8
**Confidence:** 3

**Summary:**

In this work, the authors examine sources of training instabilities in transformer models through a detailed experimental study.
They motivate their study with the fact that instabilities observed in large transformer models are difficult to study and mitigate because of the large computational costs of these runs.
They therefore examine these and show that they can be reproduced in smaller models, which can be trained faster and can be used to design mitigations for the instabilities which will hopefully translate to larger architectures.
In particular, they focus on two instabilities observed in practice, namely the the growth of logits in attention layers, and the divergence of output logits.
They show that increasing the learning rate at training time can reproduce these instabilities for smaller models.
Further, they show that commonly used mitigation approaches, such as qk-layernorm and z-regularisation can help with instabilities induced by large learning rates, and also examine the effect of a range of other optimiser and model interventions to the sensitivity of the training procedure on the learning rate.
A range of ablation studies across learning rates, interventions and model size yield a number of practical insights on training stability.

**Strengths:**

Overall, I think the experimental work in this paper was well executed and carefully controlled.
The strengths of the paper, in my view, include a number of useful findings and insights, as well as the overall high quality of the ablations and the paper itself:

__Reproducing instabilities on small models:__
The authors successfully reproduce training instabilities on smaller transformers, by increasing the learning rate.
They show that as model size increases (e.g. figure 1 and figure 6), training instabilities occur at smaller learning rates.
Furthermore, the authors show that two existing instabilities that are observed in large transformers (i.e. the growth of logits in attention layers and the divergence of output logits) can be reproduced in smaller models.
This is convincing evidence that the authors' findings on interventions made on smaller models are likely to translate to larger ones, since the mechanism of the instabilities is common across different scales.
In addition, pointing out this relationship is interesting and also potentially useful towards the development of large transformer models, as it provides strong evidence for adjusting the learning rate as a function of model size.


__Verifying the effectiveness of qk-layernorm and z-regularisation:__
The authors showed that using qk-layernorm (figure 1) and/or z-regularisation (figure 2) significantly helps mitigate instabilities, reducing sensitivity to the learning rate across a range of model sizes, and increases the range of stable learning rates.
This suggests that qk-layernorm and z-regularisation are good candidates for mitigating instabilities in small models, and likely also sufficient for mitigating these effects in large transformers as well.


__Extrapolating instabilities:__
The authors demonstrate that the hyperparameter regimes which result in instabilities can be predicted by looking at the maximum attention logits from other runs.
In particular, in figure 6, they show that for a model with no qk-layernorm, both the value of the maximum attention logit as well as the occurrence of an instability can be predicted by extrapolating from smaller runs and different learning rates.

__Overall thoroughness of ablations:__
I found that the ablations performed in this work were very thorough and supported the claims made in the main text very well.
The documentation of the various parameter settings used in the experiments are also clearly documented.

__Motivation and clarity:__
Overall, I also found the paper to be well motivated and clear, and the figures to be insightful and informative.

**Weaknesses:**

I did not find significant flaws in the paper, I thought that two possible weakness are the following:

__Absence of concrete rules of thumb:__
One weaker point in the paper is that it does not provide concrete rules of thumb for setting the relevant hyperparameters of transformer models and their training loops.
Specifically, I think that the paper goes a long way reproducing instabilities and performing detailed ablations, but does not provide concrete advice (i.e. general recipes) for hyperparameter settings.
Given the thoroughness of the ablations, this is a relatively minor point.
However, I think that a short discussion of how a practitioner could use the insights in this paper to fix training instabilities and extract better model performance (by utilising smaller scale runs), would be useful.

__Limitation to C4 data:__
To my understanding, all experiments in this work involve the C4 dataset, which is textual.
While it is most likely that the authors' findings generalise to other datasets, it is not fully clear that the scalings shown in this paper would be encountered in other data modalities.
However, I appreciate that performing experiments on additional data modalities would be a large overhead in effort, and the current findings to be convincing enough.

**Questions:**

- __Figure 1:__
The caption says "LR sensitivity measures the expected deviation from optimal."
What do the authors mean by "optimal" in this context?
Is the meaning of "optimal" coming from the discussion in section 2.2?
Some clarification on this in the main text would be good.

- __Introduction comment:__
"One interesting finding is that scaling depth increases LR sensitivity at a faster rate than scaling width."
One factor at play with this finding may be the fact that in standard initialisation schemes, changing the width of the network affects the initialisation scale of the weights, whereas increasing the depth does not.
As a result, it is reasonable to expect that changing the width does not impact stability as much as depth, because the change in width is somewhat accounted for by the adaptive initialisation.
Can the authors comment on why this occurs?

- __Point on phrasing:__
In section 3.3 the authors write "We now examine whether it is possible to predict the logit growth instability before it occurs."
I think this phrasing is a little ambiguous because it may be interpreted as predicting whether a logit growth instability will occur in an ongoing run, based on the data collected in the current run.
By contrast, to my understanding, the authors are using previous runs with different hyperparameters, to determine whether a particular hyperparameter setting will cause an instability or not.
I think stating this more clearly in the main text would be beneficial.

- __Effect of different optimisers:__
To my understanding, all experiments in this paper use AdamW.
Can the authors comment on whether they expect their findings to extend to other commonly used optimisers?

---

> ### Author Response · Authors · 2023-11-22
>
> Thank you for the comprehensive and thoughtful review. We are very glad you found the work to include a number of useful findings and insights. We hope that our comments below resolve any remaining questions:
>
> > Figure 1: The caption says "LR sensitivity measures the expected deviation from optimal." What do the authors mean by "optimal" in this context? Is the meaning of "optimal" coming from the discussion in section 2.2? Some clarification on this in the main text would be good.
>
> We apologize that this was unclear. In this context, we use optimal to refer to $\ell^*$ in Section 2.2, which is the minimum loss achieved in the learning rate sweep. To address this, we have revised the caption to read “LR sensitivity measures the expected deviation from the minimum achieved loss when varying learning rate across three orders of magnitude”.
>
> > Introduction comment: "One interesting finding is that scaling depth increases LR sensitivity at a faster rate than scaling width." One factor at play with this finding may be the fact that in standard initialisation schemes, changing the width of the network affects the initialisation scale of the weights, whereas increasing the depth does not. As a result, it is reasonable to expect that changing the width does not impact stability as much as depth, because the change in width is somewhat accounted for by the adaptive initialisation. Can the authors comment on why this occurs?
>
> We believe it is an open question why depth increases LR sensitivity at a faster rate than width in our experiments. As you suggest, a different initialization or parameterization when scaling depth may resolve this issue. This is an interesting question for future research.
>
> > Point on phrasing: In section 3.3 the authors write "We now examine whether it is possible to predict the logit growth instability before it occurs." I think this phrasing is a little ambiguous because it may be interpreted as predicting whether a logit growth instability will occur in an ongoing run, based on the data collected in the current run. By contrast, to my understanding, the authors are using previous runs with different hyperparameters, to determine whether a particular hyperparameter setting will cause an instability or not. I think stating this more clearly in the main text would be beneficial.
>
> Thank you for the suggestion. We agree and have changed the main text to state this more clearly as you’ve recommended. In our revised version, Section 3.3 begins with: “A central question when studying instabilities is whether they can be predicted using small-scale proxy experiments. We now examine whether it is possible to predict the logit growth instability before it occurs using previous runs with smaller models.”.
>
> > Effect of different optimisers: To my understanding, all experiments in this paper use AdamW. Can the authors comment on whether they expect their findings to extend to other commonly used optimisers?
>
> Indeed, we use AdamW in our experiments. We choose AdamW because it is a popular optimizer used for many large scale runs including GPT-3 and LLaMA. We believe that some of the Transformer training instabilities we study are present when using other optimizer variants. For instance, the output logit divergence instability (Section 3.1.2) is reported by Chowdhery et al., 2022 for the PaLM model, which uses the “parameter scaling” feature from Adafactor. On the other hand, we expect that some instabilities we study are absent when using other optimizers. For instance, the epsilon instability from Section 3.4 is not expected when using SGD, since SGD has no epsilon hyperparameter.
>
> > Weaknesses: “Absence of concrete rules of thumb” and “Limitation to C4 data”.
>
> We absolutely agree with these limitations. We have revised Section 5 to include a subsection which highlights both of these limitations.
>
> As you highlight in your review, “the experimental work in this paper was well executed and carefully controlled”. Unfortunately, we believe that in order to provide concrete rules of thumb or study the effect of data, a large number of further experiments would be required for us to be confident in the findings and maintain the overall standard of careful experimentation. Therefore, we believe that these directions are promising topics for further research, but beyond the scope of our investigation.

---

> > ### Comment · Reviewer_TVMA · 2023-11-22
> > **Response to rebuttal**
> >
> > Thank you for your rebuttal.
> > I am happy to hear you found the review comprehensive and thorough, and that you incorporated some of the suggestions I made.
> > Your response addressed the points that I raised well, so I will maintain the high score that I gave to the paper.

---

### Author Response · Authors · 2023-11-22

We sincerely thank all reviewers, and have uploaded a revised version of the paper based on the helpful comments we’ve received.

---

### Public Comment · ~Dongseong_Hwang1 · 2024-05-07
**AdamW does not decouple weight decay from learning rate.**

Thank you for the interesting research and congratulations on the ICLR acceptance. I have a question regarding a point in your paper that I found puzzling.

Your paper states:

"Parameterizing weight decay independently of learning rate reduces LR sensitivity, as illustrated in Figure 6. While this was recommended by Loshchilov and Hutter [33], it is not common practice in the default AdamW implementations of PyTorch [36] or Optax [2]."

However, in the AdamW paper, decoupling is suggested to be done with the loss, not the learning rate. In the paper, weight decay is still subject to the learning rate schedule. https://openreview.net/forum?id=Bkg6RiCqY7

---

> ### Public Comment · ~Jaehoon_Lee1 · 2024-05-08
> **Clarifying independent WD**
>
> Note that there is learning rate (step size) denoted as $\alpha$  and schedule $\eta_t$ in Loshchilov and Hutter [33]'s Algorithm 2. In most implementations, product of these quantities is used as learning rate schedule.
>
> In the AdamW paper (e.g. Algorithm 2) $\eta_t$ represents only schedule. For warmup + decay this is a function valued within [0, 1].  This  does not include what we often think of as learning rate (or step size which is represented by $\alpha$). Indeed AdamW suggests sharing the schedule part ($\eta_t$) for optimization update and weight decay. We are pointing out that the notion of decay strength should be independent of scale of step size (learning rate) as faithfully following the AdamW paper and not as done in common implementations.
>
> "decoupling is suggested to be done with the loss"; we believe `decoupling` is respect to optimization (gradient descent update) step and weight decay which is consistent with what we are saying in the paper.
>
> Let us know if you have any further questions.

---

### Meta-Review · Area_Chair_iD4m · 2023-12-06

**Metareview:**

This paper aims to reproduce large transformer-model training instabilities in smaller transformer models. The motivation for this work is that many transformer training instabilities only occur in sufficiently large models, making it challenging to diagnose or reproduce issues. The authors demonstrate that small models with large learning rates often replicate the instabilities of their large model counterparts, and further examine the effects common mitigation strategies and optimizer choices. Overall, this offers a highly practical solution to a very timely and relevant problem. Though the proposed method is simple, the ablation studies and experimental results thoroughly demonstrate its efficacy. I believe this paper will be of high interest to the ICLR community.

**Justification For Why Not Higher Score:**

N/A

**Justification For Why Not Lower Score:**

This paper tackles an extremely relevant problem with a practical solution. The method is simple highly efficacious. The analysis is through and well-executed. I believe this paper is exemplary of a solid and significant research contribution.

---

### Decision · Program_Chairs · 2024-01-16

Accept (oral)